# GPR174 knockdown enhances blood flow recovery in hindlimb ischemia mice model by upregulating AREG expression

Jin Liu[1,2,3,4,11], Lihong Pan[3,4,5,11], Wenxuan Hong[1,3,4,11], Siqin Chen[3,4,5,11], Peiyuan Bai[1,3,4], Wei Luo[1,3,4], Xiaolei Sun[1,3,4], Furong He[6], Xinlin Jia[7], Jialiang Cai[8], Yingjie Chen[9], Kai Hu[1], Zhenju Song [10] ✉, Junbo Ge [1,3,4,5] & Aijun Sun [1,2,3,4,5] ✉

Regulatory T cells (Tregs) are critically involved in neovascularization, an important compensatory mechanism in peripheral artery disease. The contribution of G protein coupled receptor 174 (GPR174), which is a regulator of Treg function and development, in neovascularization remains elusive. Here, we show that genetic deletion of GPR174 in Tregs potentiated blood flow recovery in mice after hindlimb ischemia. GPR174 deficiency upregulates amphiregulin (AREG) expression in Tregs, thereby enhancing endothelial cell functions and reducing pro-inflammatory macrophage polarization and endothelial cell apoptosis. Mechanically, GPR174 regulates AREG expression by inhibiting the nuclear accumulation of early growth response protein 1 (EGR1) via Gαs/cAMP/PKA signal pathway activation. Collectively, these findings demonstrate that GPR174 negatively regulates angiogenesis and vascular remodeling in response to ischemic injury and that GPR174 may be a potential molecular target for therapeutic interventions of ischemic vascular diseases.

Peripheral artery disease (PAD) affects more than 200 million individuals worldwide and manifests as intermittent claudication or critical limb ischemia (CLI) characterized by ischemic rest pain, ulceration, and gangrene, potentially leading to amputation and early death[1,2]. Common triggers of PAD include atherosclerotic vascular occlusions, metabolic syndrome, and diabetes mellitus[3–5]. PAD is also associated with a high risk of cardiovascular incidents, including stroke and myocardial infarction[6]. Although the pathological process has been well studied, there are few effective treatment options to improve blood flow in ischemic tissues. Therefore, additional studies are warranted to explore novel approaches to ameliorate ischemic tissue injury and improve perfusion.

Accumulating evidence has demonstrated the infiltration of Foxp3[+] regulatory T cells (Tregs) into injured tissues, including cardiac and skeletal muscle, the lungs, skin, and brain, promoted vascular and tissue regeneration as well as wound healing[7–11]. Tregs have also been reported to enhance the regeneration of peripheral vasculature in PAD[12,13]. Conversely, other studies have revealed that depletion of

[1]Department of Cardiology, Shanghai Institute of Cardiovascular Diseases, Zhongshan Hospital, Fudan University, Shanghai 200032, China. [2]Human Phenome Institute, Fudan University, 825 Zhangheng Road, Shanghai 201203, China. [3]Key Laboratory of Viral Heart Diseases, National Health Commission, Shanghai 200032, China. [4]Key Laboratory of Viral Heart Diseases, Chinese Academy of Medical Sciences, Shanghai 200032, China. [5]Institutes of Biomedical Sciences, Fudan University, Shanghai 200032, China. [6]Department of Endocrinology, Xiang'an Hospital of Xiamen University, Xiamen, Fujian 361000, China. [7]Shanghai Key Laboratory of Orthopaedic Implants, Department of Orthopaedic Surgery, Shanghai Ninth People's Hospital, Shanghai Jiao Tong University School of Medicine, Shanghai 200125, China. [8]Liver Cancer Institute, Zhongshan Hospital, Fudan University, Shanghai 200032, China. [9]Department of Physiology and Biophysics, University of Mississippi Medical Center 2500N. State St, Jackson, MS 39216-4505, USA. [10]Department of Emergency Medicine, Zhongshan Hospital, Fudan University, Shanghai 200032, China. [11]These authors contributed equally: Jin Liu, Lihong Pan, Wenxuan Hong, Siqin Chen. ✉e-mail: song.zhenju@zs-hospital.sh.cn; sun.aijun@zs-hospital.sh.cn

Tregs exhibited increased restoration of blood flow after HLI[14,15]. However, the molecular mechanisms by which Tregs regulate angiogenesis and vascular remodeling in ischemic tissues are not fully understood.

GPR174, an X-linked G protein coupled receptor (GPCR) that belongs to the P2Y receptor family, is abundantly expressed in immune cells, particularly B and T lymphocytes[16]. Recently, investigations have identified the bioactive lipid lysophosphatidylserine (LysoPS) as a high-affinity GPR174 ligand that suppresses Treg homeostasis and proliferation[17]. There are also multiple lines of evidence suggesting that, GPR174 couples to Gαs to regulate the cyclic AMP/protein kinase A (cAMP/PKA) signaling pathway, which then suppresses the proliferation and function of T cells[18,19]. However, whether GPR174 affects the progression of ischemic diseases by regulating Treg function remains to be elucidated.

Amphiregulin (AREG), a member of the epidermal growth factor (EGF) family, has been shown to play an important role in wound healing and tissue repair[20]. Recent studies have suggested that AREG produced by Tregs in response to injury and stress promotes tissue regeneration[7–11]. The increased production of AREG by Tregs limits the polarization of pro-inflammatory macrophages and in turn promotes tissue repair[21]. AREG has also been reported to be involved in angiogenesis, proliferation, and apoptosis and to prevent myocardial ischemia-reperfusion injury[22–24]. However, the effects of AREG in PAD remain unclear.

In the present study, we investigate the role of GPR174 in blood flow recovery using a hindlimb ischemia (HLI) mouse model. We demonstrate that GPR174 deficiency in Tregs mitigates the inflammatory response and improves endothelial cell proliferation and survival by promoting AREG secretion. Furthermore, we uncover early growth response protein 1 (EGR1) as a transcription factor that elevates AREG expression. We also confirm that GPR174 interacts with Gαs to inhibit the nuclear localization of EGR1 by triggering cAMP/PKA activity. Combined, our data demonstrate the crucial role of GPR174 in regulating inflammation and endothelial cell survival after ischemic injury and identify Treg GPR174 as a potential therapeutic target for treating ischemic vascular diseases, such as PAD.

## Results

### GPR174 knockout enhances blood flow recovery in HLI mice
To determine the possible role of GPR174 in blood flow recovery, we used Laser Doppler imaging to evaluate blood flow in wild-type (WT) and GPR174-knockout (Gpr174−/Y) mice after HLI (Supplementary Fig. 1a, b). The baseline muscle vascular density was similar between in Gpr174−/Y and WT mice (Supplementary Fig. 1c, d). However, Gpr174−/Y mice displayed faster blood flow recovery at 7 days after HLI compared with littermate controls (Fig. 1a, b). Moreover, Gpr174−/Y mice exhibited a lower frequency of necrotic toes (Fig. 1c). Because arteriogenesis and angiogenesis are involved in perfusion recovery after ischemia[25], CD31 immunostaining as an index of capillary formation was used to evaluate angiogenesis. Consistent with the Laser Doppler imaging results, vascular density was higher in Gpr174−/Y mice compared with littermate controls 7 days after HLI (Fig. 1d, e). Next, arteriogenesis was analyzed by immunofluorescence staining of vessels. Similar lumen circumferences were observed in the adductor muscles of wild-type and Gpr174−/Y mice on their non-ligated sides (Supplementary Fig. 1e, f). In contrast, lumen sizes in the ischemic adductor muscles were larger in Gpr174−/Y mice than littermate controls, indicating increased arteriogenesis post HLI in Gpr174−/Y mice (Fig. 1f, g). To further confirm this phenotype, CD45−CD31+ endothelial cells were quantified by flow cytometry. As expected, the number of endothelial cells was increased in Gpr174−/Y mice 14 days after HLI (Fig. 1h, i). The hypoxic areas in the ischemic muscles were detected by intraperitoneal injection of pimonidazole 7 days after HLI. Degree

of hypoxia was reduced in Gpr174−/Y mice, which was consistent with the increased arteriogenesis (Fig. 1j, k). In addition, the Matrigel plug assay, an ex vivo model, was used to further explore the potential impact of GPR174 deletion in angiogenesis. Gpr174−/Y mice exhibited increases in vessel formation and hemoglobin content, indicating mature vascular formation and perfusion (Fig. 1l, n). These findings were consistent with the increased neovascularization observed in Gpr174−/Y mice after HLI, providing strong evidence that GPR174 knockout promoted neovascularization. These data suggest that GPR174 plays a crucial role in ischemia-induced angiogenesis and arteriogenesis.

### GPR174-deficient Tregs improve blood flow recovery after ischemia
We next investigated the cellular source that mediates the GPR174-related angiogenic effects after HLI. Macrophages, neutrophils, ILC2s and lymphocytes, including B cells, CD8+ T cells, CD4+Foxp3− T cells, and CD4+Foxp3+ Tregs, were sorted by flow cytometry from the muscle tissues of Foxp3GFP mice, in which GFP is driven by the Foxp3 promoter, 7 days after HLI (Supplementary Fig. 2a)[26]. The expression of Gpr174 mRNA has no difference in macrophages, neutrophils, ILC2s, B cells, CD8+ T cells, and CD4+Foxp3− T cells isolated from the ischemic muscles compared with these cells isolated from the non-ischemic muscles (Fig. 2a). However, higher Gpr174 expression was detected in Tregs isolated from the ischemic areas than cells sorted from the non-ischemic areas and immunofluorescence staining showed that GPR174 was mainly expressed on the cell membranes of Tregs (Fig. 2a, b). These results prompted us to speculate that Treg-expressed GPR174 may play a crucial role in blood flow recovery after HLI. Therefore, adoptive transplantation experiment was used to further explore the functional role of GPR174-deficient Tregs in ischemia-induced neovascularization. Tregs were isolated from the spleens of wild type and Gpr174−/Y mice and were then separately injected into immunodeficient Rag1−/− mice calf muscles 3 days after HLI. There was no difference between the number of wild-type and GPR174-deficient Tregs present in the injured gastrocnemius tissues of recipients (Supplementary Fig. 2b, c). The mice treated with either wild-type or GPR174-deficient Tregs exhibited improved blood flow recovery after HLI compared with the PBS-injected controls. However, the effects of GPR174-deficient Tregs were more pronounced (Fig. 2c–e). Immunofluorescence staining of the ischemic tissues showed that adoptive transfer with GPR174-deficient Tregs increased vascular density (Fig. 2f, g). Additionally, increased artery lumen diameters were observed in the injured adductor muscles of Rag1−/− mice injected with GPR174-deficient Tregs (Fig. 2f, g). These data suggest that Treg-expressed GPR174 negatively regulates blood flow recovery after ischemia.

To further confirm the role of Treg-expressed GPR174 in neovascularization, tube formation and aortic ring assay were performed. Tregs were isolated from the ischemic muscles of wild-type and Gpr174−/Y mice and then were cocultured with mouse aortic endothelial cells (MAECs) or the aortic segments in transwell inserts (Fig. 2h). As expected, elevated tube formation in MAECs and vascular sprouting in the aortic segments were observed in coculture with GPR174-deficient Tregs (Fig. 2i–l).

### Lysophosphatidylserine is an endogenous ligand of GPR174
Previous studies have demonstrated that lysophosphatidylserine (LysoPS) and C-C motif chemokine ligand 21 (CCL21) function as endogenous ligands of GPR174 to activate the classical G protein-mediated signaling[16,27]. Therefore, mass spectrometry (MS) and enzyme-linked immunosorbent assay (ELISA) were performed to determine the expression of these ligands in wild-type mice after HLI. Results showed that 18:0 LysoPS production was elevated, whereas CCL21 expression remained unchanged (Supplementary Fig. 3a, b).

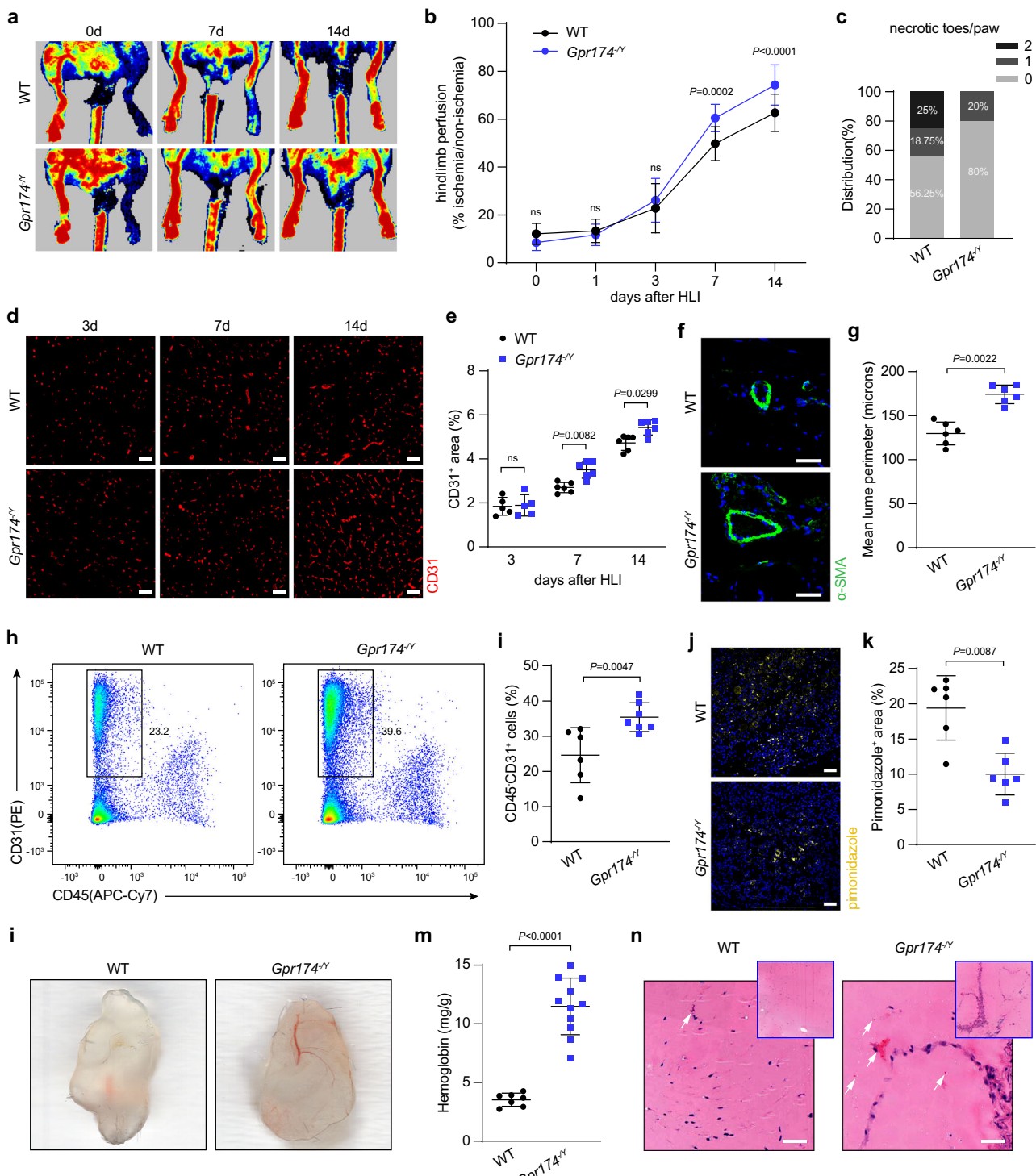

**Fig. 1 | GPR174 deficiency improves blood flow recovery after HLI.**
**a**, **b** Representative images and quantification of hindlimb blood perfusion in WT and *Gpr174⁻/ᵞ* mice using Laser Doppler imaging at the indicated times after HLI (*n* = 16 for WT mice; *n* = 15 for *Gpr174⁻/ᵞ* mice). **c** Distribution of necrotic toes per paw 14 days after HLI (*n* = 16 for WT mice; *n* = 15 for *Gpr174⁻/ᵞ* mice). **d**, **e** Representative immunofluorescent images of CD31 staining and quantification of CD31⁺ area in WT and *Gpr174⁻/ᵞ* mice gastrocnemius cross sections at the indicated times after HLI (*n* = 5 for WT and *Gpr174⁻/ᵞ* mice 3 days after HLI; *n* = 6 for WT and *Gpr174⁻/ᵞ* mice 7 and 14 days after HLI). Scale bar, 50 μm. **f**, **g** Representative images of αSMA (green) and DAPI (blue) immunostainings and quantification of lumen perimeter in WT and *Gpr174⁻/ᵞ* mice adductor cross sections 14 days after HLI (*n* = 6; wild-type: 50 arteries from 6 mice, *Gpr174⁻/ᵞ*: 51 arteries from 6 mice). Scale

bar, 50 μm. **h**, **i** Quantification of CD45⁻CD31⁺ endothelial cells in WT and *Gpr174⁻/ᵞ* mice muscle using Flow cytometry 14 days after HLI (*n* = 6 for WT mice; *n* = 7 for *Gpr174⁻/ᵞ*). **j**, **k** Representative images of pimonidazole (yellow) and DAPI (blue) immunostainings and quantification of hypoxic areas at 7 days (*n* = 6). Scale bar, 50 μm. **l** Representative images of neovascularization in Matrigel plugs ex vivo in WT and *Gpr174⁻/ᵞ* mice. **m** Quantification of hemoglobin in Matrigel plugs in WT and *Gpr174⁻/ᵞ* mice (*n* = 7 for WT mice; *n* = 11 for *Gpr174⁻/ᵞ* mice). **n** Representative hematoxylin-eosin (H&E) staining images of Matrigel plugs. Scale bar, 50 μm. For all statistical plots, the data are presented as mean ± SD. Two-way repeated measures ANOVA with Sidak's multiple comparisons test in (**b**). One-way ANOVA with Bonferroni multiple comparisons test in (**e**). Two-tailed unpaired t-tests in (**g**, **i**, **k**, **m**). Source data are provided as a Source Data file.

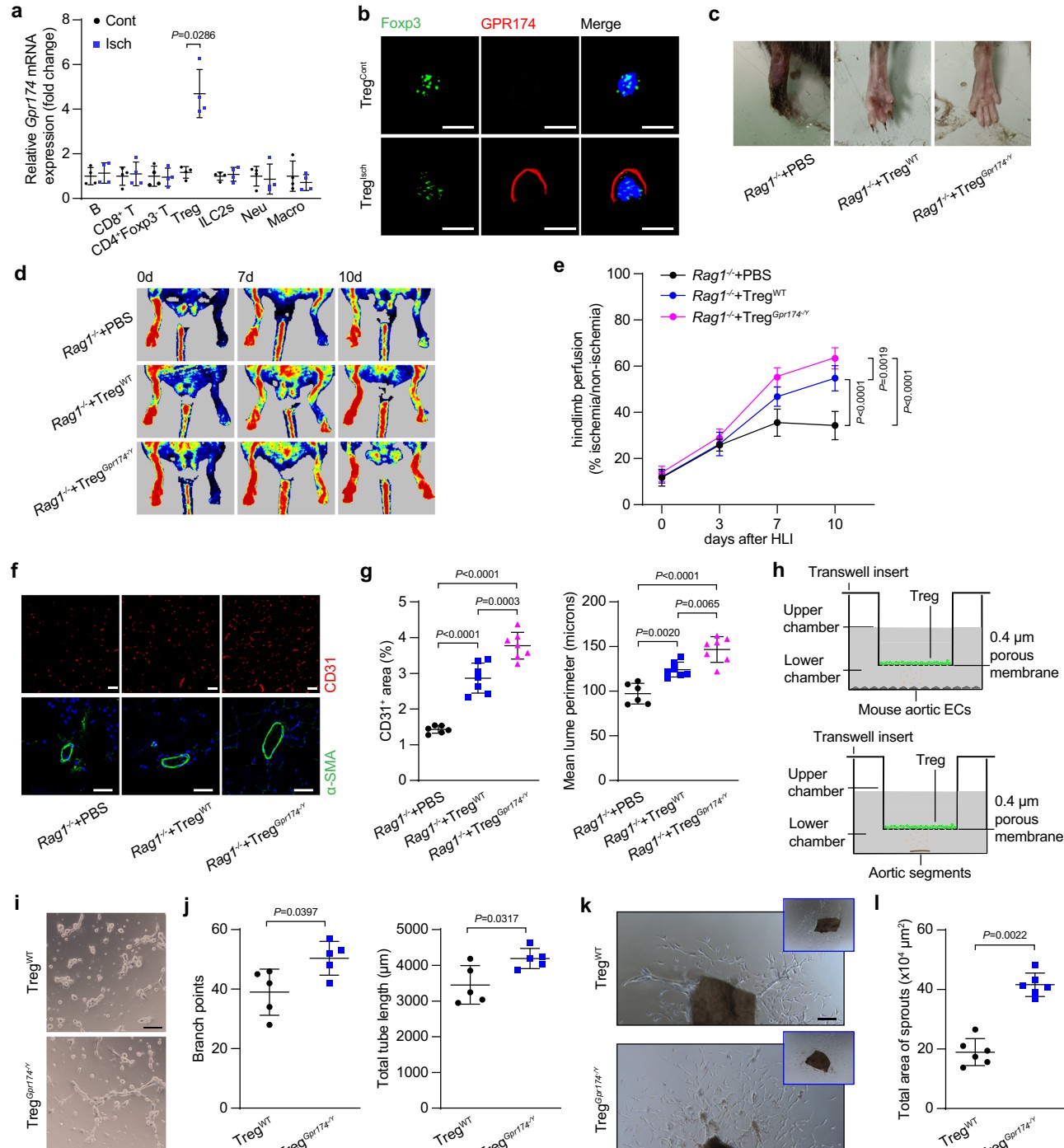

**Fig. 2 | GPR174-deficient Tregs improve blood flow recovery after HLI. a** *Gpr174* mRNA expression in B cells, CD8⁺ T cells, CD4⁺Foxp3⁻ T cells, CD4⁺Foxp3^GFP Tregs, ILC2s, neutrophils, and macrophages isolated from non-ischemic and ischemic muscle in WT mice (*n* = 4). **b** Representative immunofluorescent images of GPR174 (red), Foxp3 (green), and DAPI (blue) in CD4⁺Foxp3^GFP Tregs isolated from non-ischemic and ischemic muscle in WT mice. Scale bar, 10 µm. **c** Representative images of necrotic limb 7 days after Tregs adoptive transfer. **d, e** Representative Laser Doppler images and quantification of hindlimb blood perfusion in *Rag1*⁻/⁻ mice upon WT or GPR174-deficient Tregs (*n* = 6 for PBS → *Rag1*⁻/⁻ mice; *n* = 7 for wild-type Tregs→*Rag1*⁻/⁻ mice; *n* = 7 for GPR174-deficient Tregs→*Rag1*⁻/⁻ mice). **f** Representative immunofluorescent images of CD31(red) (upper panels) and αSMA (green) (lower panels) and DAPI (blue) staining in ischemic muscle cross sections. Scale bar, 50 µm. **g** Quantification of CD31 (left) and lumen perimeter

(right) in *Rag1*⁻/⁻ mice upon WT or GPR174-deficient Tregs (*n* = 6 for PBS → *Rag1*⁻/⁻ mice: 37 arteries from 6 mice; *n* = 7 for wild-type Tregs→*Rag1*⁻/⁻ mice: 46 arteries from 7 mice; *n* = 7 for GPR174-deficient Tregs→*Rag1*⁻/⁻ mice: 47 arteries from 7 mice). **h** Scheme of mouse aortic endothelial cell tube formation and aortic ring assay. **i, j** Representative images of capillary-like structures and quantification of branch points and total tube length in mouse aortic endothelial cells cocultured with Tregs isolated from WT or *Gpr174*⁻/Y mice for 16 h (*n* = 5). Scale bar, 200 µm. **k, l** Representative images and quantification of vascular sprouting of vessel segments cocultured with Tregs isolated from WT or *Gpr174*⁻/Y mice for 5 days (*n* = 6). Scale bar, 200 µm. For all statistical plots, the data are presented as mean ± SD. Two-tailed unpaired t-tests in (**a, j, l**). Two-way repeated measures ANOVA with Sidak's multiple comparisons test in (**e**). One-way ANOVA with Bonferroni multiple comparisons test in (**g**). Source data are provided as a Source Data file.

These findings indicate that 18:0 LysoPS might serve as the endogenous ligand of GPR174 after HLI.

## Amphiregulin expression is upregulated in GPR174-deficient Tregs

To further identify the downstream targets of GPR174 in response to HLI, RNA sequencing (RNA-seq) analysis of the ischemic muscles from *Gpr174*[−/Y] mice and littermate controls was performed 7 days after HLI. Analysis of the genome-wide gene expression profiles revealed that *Amphiregulin* (*Areg*) expression was increased in *Gpr174*[−/Y] mice (Fig. 3a; Supplementary Fig. 4a, b). Moreover, previous study has shown that Tregs within injured muscles express high levels of *Areg*[7] (Supplementary Fig. 4c). We thus speculated that GPR174 may regulate AREG expression in Tregs after ischemic injury. RT-qPCR confirmed the upregulation of *Areg* and *Vegf* in the ischemic muscles in *Gpr174*[−/Y] mice 7 days after HLI as compared with wild-type controls, but there was no difference in *Il-10* expression (Supplementary Fig. 5a). Based on ELISA experiments, serum AREG and VEGF concentrations were higher in *Gpr174*[−/Y] mice than littermate controls (Supplementary Fig. 5b). Moreover, in comparison to *Rag1*[−/−] mice receiving wild-type Tregs, serum AREG and VEGF levels were elevated in those receiving GPR174-deficient Tregs at 7 days after adoptive transplantation (Fig. 3b). We further examined the expression of *Areg*, *Vegf*, and *Il-10* in Tregs isolated from the muscle tissues of *Gpr174*[−/Y] and wild-type mice 7 days after HLI. The results demonstrated that, in comparison to wild-type Tregs, GPR174-deficient Tregs from the ischemic muscles exhibited higher *Areg* expression (Fig. 3c). However, there was no difference in the expression of *Vegf* and *Il-10* in these cells (Fig. 3c). Immunofluorescence staining also confirmed increased AREG expression in GPR174-deficient Tregs (Fig. 3d).

Based on these findings, we further examined whether GPR174-deficient Treg-induced neovascularization could be inhibited by an exogenous AREG neutralizing antibody. In accordance with such a hypothesis, aortic ring assay showed that the total area of vascular sprouting of vessel segments cocultured with GPR174-deficient Tregs for 5 days was reduced in the presence of exogenous AREG neutralizing antibody (Fig. 3e, f; Supplementary Fig. 5c).

To further explore the role of AREG on GPR174-mediated neovascularization post ischemia in vivo, we knocked down AREG in mice using an adeno-associated virus (AAV9) bearing sh*RNA* against *Areg* (Supplementary Fig. 5d). Hindlimb perfusion measurements showed that both wild-type and *Gpr174*[−/Y] mice exhibited reduced hindlimb reperfusion and disrupted neovascularization after injection with AAV9-sh*Areg* (Fig. 3g–l).

Given the increased angiogenesis in Matrigel plugs, Tregs infiltration and AREG levels were measured. The results showed that Tregs infiltration in Matrigel plugs had no difference in between wild-type control and GPR174-deficient group, but AREG levels were increased in the latter group (Supplementary Fig. 6a, b).

Collectively, these data suggest that GPR174 deficiency enhanced ischemia-induced revascularization at least partially by upregulating AREG expression.

## AREG inhibits apoptosis and promotes proliferation in endothelial cell

Immunostaining was performed in the ischemic muscles of *Rag1*[−/−] mice upon transfer of Tregs to evaluate endothelial cell functions. As expected, more CD31[+]Ki67[+] cells were found in the gastrocnemius tissues of recipients receiving GPR174-deficient Tregs than recipients receiving wild-type Tregs (Fig. 4a). Transfer of GPR174-deficient Tregs also reduced CD31[+]TUNEL[+] cells in the ischemic muscles of *Rag1*[−/−] mice (Fig. 4b). This observation also coincided with an increased vascular density in these mice. Perfusion recovery in *Gpr174*[−/Y] vs. controls might already be different early after HLI. To address this issue, we assessed reperfusion and angiogenesis at

3 days post HLI. Results showed reperfusion and angiogenesis were similar between WT and *Gpr174*[−/Y] mice at 3 days after HLI (Fig. 1a, b, d, e). Further, immunofluorescence was performed to detect the proliferation and apoptosis of ECs after HLI. Elevated proliferation and reduced apoptosis of ECs were evidenced in GPR174 KO mice at 3 days after HLI (Supplementary Fig. 7a, b), suggesting that GPR174 knockout promotes blood flow recovery by affecting endothelial cell proliferation and apoptosis.

Given the crucial role of AREG in proliferation and apoptosis[23], we next evaluated the effect of AREG on endothelial cell functions. Results showed that the effect of GPR174 deficiency was partly blocked by AREG knockdown, indicated by reduced proliferation and enhanced apoptosis in the ischemic muscles (Fig. 4c, d). These observations also coincided with an increased vascular density in these mice. Further, tube formation was performed and elevated total tube length was observed in MAECs stimulated with recombinant AREG compared with control groups (Fig. 4e). Moreover, tube formation was also enhanced in human umbilical vein endothelial cells (HUVECs) treated with recombinant AREG (Supplementary Fig. 8a, b). Immunoblotting analysis demonstrated that the expression of the pro-apoptotic molecules, cleaved caspase-3 and BCL-2-associated X protein (BAX), was decreased and that the expression of the anti-apoptotic molecule, B-cell lymphoma 2 (BCL-2), was increased in HUVECs treated with recombinant AREG (Supplementary Fig. 8c, d). To further confirm the effects of AREG on endothelial cell apoptosis, we labeled the apoptotic endothelial cells with annexin V. Consistent with the above findings, endothelial cell apoptosis was reduced following recombinant AREG treatment. However, this effect was not apparent during early apoptosis (Supplementary Fig. 8e, f).

We next investigated the effects of AREG on endothelial cell adherens junctions due to their importance in vascular permeability and angiogenesis. Western blot analysis revealed increased expression of endothelial cell adhesion molecule, VE-cadherin, in HUVECs after recombinant AREG treatment (Supplementary Fig. 8g). Immunofluorescence staining further showed enhanced adherens junctions between endothelial cells stimulated with recombinant AREG (Supplementary Fig. 8h). These data suggest that AREG enhances angiogenesis at least partially by protecting against endothelial cell apoptosis and promoting endothelial cell proliferation and cell-cell interactions.

## GPR174-deficient Tregs attenuate the inflammatory phenotypic transformation of macrophages in ischemic tissues

Previous study demonstrated that GPR174-deficient Tregs mitigated lung injury by promoting macrophages polarization[28], and macrophages also play a critical role during skeletal muscle regeneration after injury[29]. To further investigate whether GPR174-deficient Tregs affect immune cell recruitment/infiltration and macrophage polarization, flow cytometry was performed to evaluate the number of various myeloid cell populations in the gastrocnemius tissues of *Gpr174*[−/Y] and wild-type mice at 7 days after HLI. Gating strategies were used and CD45[+]CD11b[+]Ly6G[+]F4/80[−] cells were defined as neutrophils and CD45[+]CD11b[+]Ly6G[−]F4/80[+] cells as the total muscle monocyte/macrophage population. The CD45[+]CD11b[+]Ly6G[−]F4/80[+] cells were further divided into two subsets based on either low or high Ly6C expression (Ly6C[low] and Ly6C[high], respectively) (Supplementary Fig. 9a). Myeloid cell populations in the non-ligation side muscles from *Gpr174*[−/Y] and wild-type mice were similar (Supplementary Fig. 9b, c). As expected, increased myeloid cell infiltration was observed in the ischemic muscles of both *Gpr174*[−/Y] and wild-type mice compared with the respective nonischemic muscles, but the infiltration of leukocyte, neutrophil, and total macrophage populations was similar between the two groups (Fig. 5a, b). In contrast, fewer pro-inflammatory Ly6C[high] macrophages and more anti-inflammatory Ly6C[low] macrophages were found in the ischemic muscles of *Gpr174*[−/Y] mice (Fig. 5a, b).

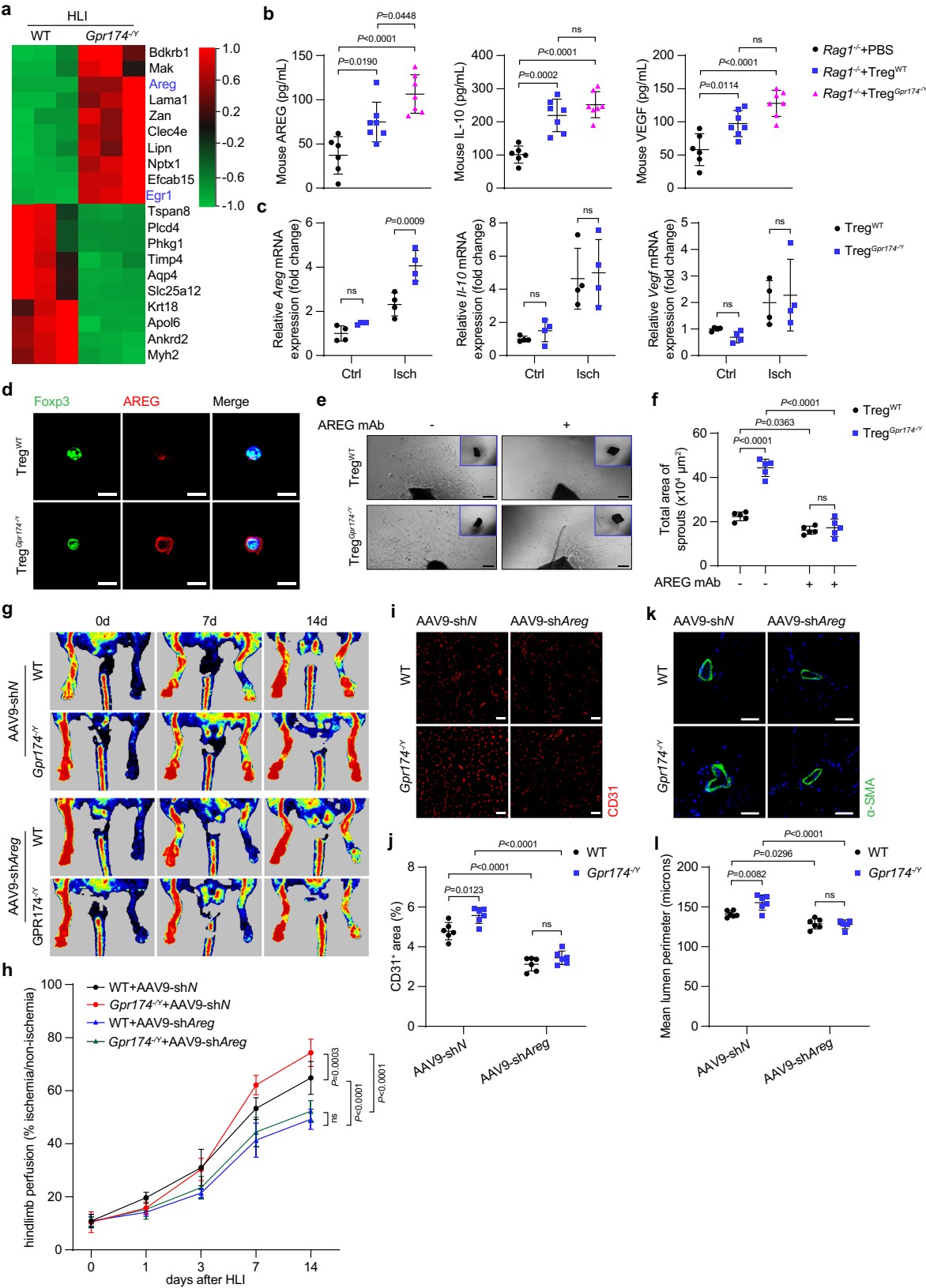

Next, coculture experiments were performed using Tregs isolated from ischemic muscle 7 days after HLI to evaluate the expression of inflammation-related genes in bone marrow-derived macrophages (BMDMs) (Fig. 5c). We found that coculture with GPR174-deficient Tregs decreased the activation of several pro-inflammatory genes in the BMDMs (Fig. 5d). Moreover, GPR174-deficient Tregs also upregulated the expression levels of anti-inflammatory genes in the BMDMs (Fig. 5d). To verify the central role of AREG on GPR174-mediated phenotypic transformation of the BMDMs, AREG neutralizing antibodies were applied and results showed that AREG neutralizing antibodies prevented the phenotypic transformation of the BMDMs induced by GPR174-deficient Tregs (Fig. 5d). Collectively, these data

**Fig. 3 | GPR174 regulates neovascularization by inhibiting AREG expression in Tregs. a** Heatmap of RNA-seq analysis of the ischemic muscles of WT and *Gpr174⁻/Y* mice 7 days after HLI (*n* = 3) (*q*-value < 0.05; |log₂Fc| > 1). **b** Serum AREG, IL-10, and VEGF protein content in *Rag1⁻/⁻* mice receiving Tregs 7 days after adoptive transplantation experiments (*n* = 6 for PBS → *Rag1⁻/⁻* mice; *n* = 7 for wild-type Tregs→ *Rag1⁻/⁻* mice; *n* = 7 for GPR174-deficient Tregs→*Rag1⁻/⁻* mice). **c** Relative mRNA levels of *Areg*, *Il-10*, and *Vegf* in Tregs sorted from the gastrocnemius tissues of WT and *Gpr174⁻/Y* mice 7 days after HLI (*n* = 4). **d** Representative immunofluorescent images of AREG (red) and DAPI (blue) staining in Tregs isolated from ischemic muscle of WT and *Gpr174⁻/Y* mice 7 days after HLI. Scale bar, 10 μm. **e**, **f** Representative images and quantification of vascular sprouting of vessel segments cocultured with AREG neutralizing antibody and Tregs isolated from WT and *Gpr174⁻/Y* mice for 5 days (*n* = 5). Scale bar, 200 μm. **g**, **h** Representative Laser Doppler images and

quantification of hindlimb blood perfusion in WT and *Gpr174⁻/Y* mice injected with AAV9-sh*Areg* at indicated times after HLI (*n* = 7 for WT mice + AAV9-sh*N* and *Gpr174⁻/Y* mice + AAV9-sh*N*; *n* = 8 for WT mice + AAV9-sh*Areg* and *Gpr174⁻/Y* mice + AAV9-sh*Areg*). **i**, **j** Representative immunofluorescent images of CD31 (**i**) staining and quantification of CD31 (**j**) in muscle cross sections (*n* = 6). Scale bar, 50 μm. **k**, **l** Representative immunofluorescent images of αSMA (**k**) staining and quantification of lumen perimeter (**l**) in muscle cross sections (*n* = 6). Scale bar, 50 μm. For all statistical plots, the data are presented as mean ± SD. One-way ANOVA with Bonferroni multiple comparisons test in (**b**). Two-way ANOVA with Bonferroni multiple comparisons test in (**c**, **f**, **j**, **l**). Two-way repeated measures ANOVA with Sidak's multiple comparisons test in (**h**). Source data are provided as a Source Data file.

demonstrate that GPR174 deficiency in Tregs improves blood flow recovery after HLI at least partially by enhancing AREG-induced macrophage polarization.

AREG has been reported to promote vascular endothelial growth factor (VEGF) production and the release of bioactive transforming growth factor beta (TGF-β)[22,30]. Next, we treated BMDMs with recombinant AREG and examined the levels of its downstream effector molecules, bioactive TGF-β and VEGF, in the supernatant. The BMDMs treated with recombinant AREG released higher levels of both bioactive TGF-β and VEGF (Fig. 5e). However, these effects were prevented in the presence of EGFR inhibitor Gefitinib (Fig. 5e).

Combined, these data reveal a mechanism by which GPR174 promotes inflammation in ischemic muscles by repressing the secretion of AREG by Tregs and in turn TGF-β and VEGF by macrophages.

## GPR174 downregulates AREG expression by inhibiting EGR1 nuclear accumulation

To further investigate the molecular mechanism by which GPR174 regulates AREG expression, RNA-seq analysis and Ingenuity Pathway Analysis (IPA; http://www.ingenuity.com/) were used to identify transcription factors that may possibly bind to *Areg* promoter. The results revealed that early growth response 1 (EGR1), a known transcription factor of growth factor, was a potential transcription factor of AREG (Figs. 3a, 6a). Previous investigations have revealed that GPR174 couples to Gαs to increase cAMP levels and PKA activity, which in turn inhibits Treg proliferation and function[18]. Furthermore, PKA has been reported to repress EGR1 nuclear localization[31]. We thus hypothesized that GPR174 might regulate AREG expression by modulating EGR1 nuclear localization. Therefore, RT-qPCR was performed to evaluate *Egr1* expression in Tregs after HLI. Although there were no differences at baseline, *Egr1* mRNA expression levels were elevated in GPR174-deficient Tregs isolated from the ischemic muscles compared with wild-type Tregs (Fig. 6b). Furthermore, immunofluorescence staining was used to analyze EGR1 nuclear localization in the Tregs. GPR174-deficient Tregs from the ischemic areas exhibited more abundant EGR1 nuclear localization than wild-type Tregs (Fig. 6c, d). These data suggest that GPR174 regulates both the expression and nuclear accumulation of EGR1 in Tregs.

We further explore the effect of GPR174 on EGR1-mediated AREG regulation. To simulate the ischemic environment in vitro, Tregs isolated from spleen of *Gpr174⁻/Y* and wild-type mice were treated with sh*Egr1* (Supplementary Fig. 10) and ischemic muscle lysates. We found that EGR1 knockdown reduced AREG expression induced by GPR174 knockout (Fig. 6e). We then investigated how LysoPS regulates AREG expression. AREG production was suppressed in Tregs treated with LysoPS and the cell-permeable cAMP analog db-cAMP (Supplementary Fig. 11a, b). In contrast, blocking cAMP-dependent PKA with Rp-cAMPS or H 89 reversed the effects of LysoPS (Supplementary Fig. 11a, b).

These data suggest that GPR174 negatively regulates AREG expression by inhibiting EGR1 nuclear accumulation.

## EGR1 activates *Areg* transcriptional activity

EGR1 was predicted as a potential transcription factor of *Areg* by IPA. Therefore, we speculated that EGR1 may bind to the *Areg* promoter to activate its transcription. The JASPAR database identified three predicted EGR1 binding sites within the *Areg* promoter (Fig. 7a, b). The first and third binding sites were located 72−85 bp and 78−91 bp, respectively, upstream of the transcription start site, whereas the second binding site was located 54−67 bp downstream of the transcription initiation site (Fig. 7b). As expected, chromatin immunoprecipitation (ChIP) and qPCR analyses revealed a higher enrichment efficiency of the *Areg* promoter with the EGR1 antibody than the normal anti-IgG antibody (Fig. 7c; Supplementary Fig. 12a), indicating EGR1 as a transcriptional regulator of *Areg*. After stimulation with 18:0 LysoPS, the enrichment of EGR1-bound *Areg* promoter was reduced by CRISPR activation plasmid-mediated GPR174 overexpression in HEK293A cells (Fig. 7d; Supplementary Fig. 12b). These data indicate that GPR174 signaling affects the binding of EGR1 to the promoter regions of *Areg*.

To further determine the mechanism of EGR1-mediated *Areg* promoter activation, we mutated the three EGR1 binding sites in the *Areg* promoter and individually transfected the respective plasmids containing either mutated or wild-type promoter regions into HEK293A cells overexpressing EGR1. We found decreased *Areg* promoter activity in the HEK293A cells expressing either the first or third mutated ERG1 binding site compared with those expressing the wild-type binding sites (Fig. 7e, f; Supplementary Fig. 12c, d). We further examined whether GPR174 regulated *Areg* transcription via the PKA/EGR1 pathway. After stimulation with 18:0 LysoPS, GPR174 overexpression attenuated *Areg* promoter activity, whereas the promoter activity was enhanced in response to PKA inhibition (Fig. 7g). Altogether, these data indicate that GPR174 regulates *Areg* transcriptional activation via the PKA/EGR1 pathway (Supplementary Fig. 12e).

## GPR174 knockout enhances blood flow recovery after HLI in diabetic mice

Diabetes mellitus and metabolic syndrome are involved in the development of PAD. We next investigated the effects of GPR174 using the streptozotocin (STZ)-induced type I diabetic mouse model. Hindlimb ischemia was established on the left hind limb of mice 12 weeks after STZ intraperitoneal injections, and foot perfusion was evaluated at indicated time points after HLI (Supplementary Fig. 13a). GPR174 deletion promoted blood flow recovery angiogenesis after HLI in diabetic mice compared to wild-type diabetic mice; however, there was no difference 3 days after HLI (Supplementary Fig. 13b−e). Furthermore, immunofluorescence was also performed to evaluate the effect of GPR174 in endothelial cell proliferation and apoptosis in diabetic ischemic muscle 3 days after HLI. The results showed that endothelial cell proliferation increased and apoptosis decreased in *Gpr174⁻/Y* diabetic mice, which might partially be associated to the observed promoted perfusion recovery in later days post HLI (Supplementary

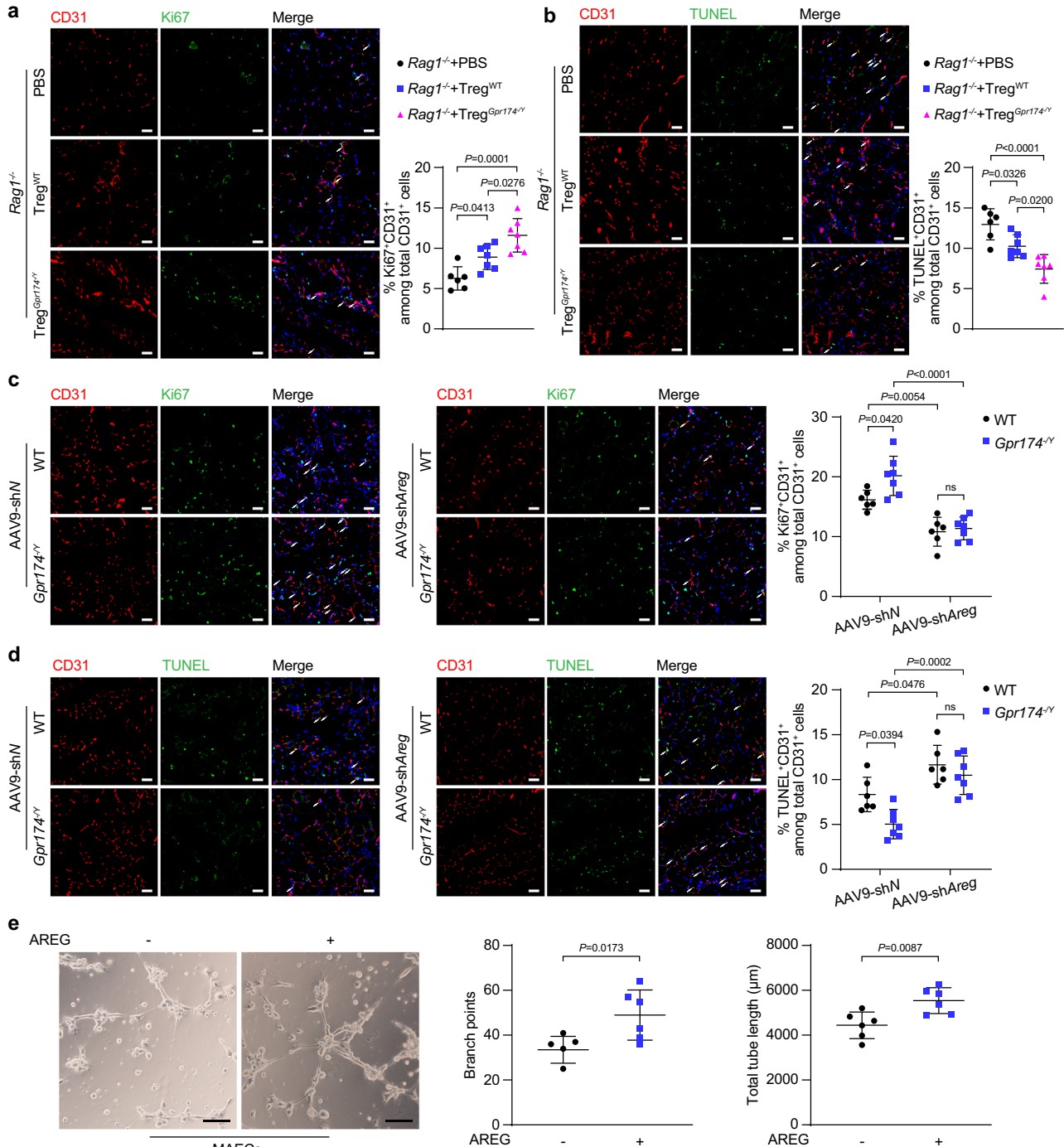

**Fig. 4 | AREG mitigates apoptosis and enhances tube formation in endothelial cells. a** Representative immunofluorescent image of CD31 (red), Ki67 (green) and DAPI (blue) and quantification of CD31⁺Ki67⁺ cells in muscle cross sections of *Rag1⁻/⁻* mice receiving Tregs 7 days after adoptive transplantation experiments (*n* = 6 for PBS → *Rag1⁻/⁻* mice; *n* = 7 for wild-type Tregs→*Rag1⁻/⁻* mice; *n* = 7 for GPR174-deficient Tregs→*Rag1⁻/⁻* mice). Scale bar, 50 μm. **b** Representative immunofluorescent images of CD31 (red), TUNEL (green), and DAPI (blue) staining and quantification of CD31⁺TUNEL⁺ cells in muscle cross sections of *Rag1⁻/⁻* mice receiving Tregs 7 days after adoptive transplantation experiments (*n* = 6 for PBS → *Rag1⁻/⁻* mice; *n* = 7 for wild-type Tregs→*Rag1⁻/⁻* mice; *n* = 7 for GPR174-deficient Tregs→*Rag1⁻/⁻* mice). Scale bar, 50 μm. **c** Representative immunofluorescent images of CD31 (red), Ki67 (green) and DAPI (blue) and quantification of CD31⁺Ki67⁺ cells in muscle cross sections of WT and *Gpr174⁻/Y* mice injection with AAV9-sh*Areg* 14 days after HLI (*n* = 6 for

WT mice + AAV9-sh*N* and WT mice + AAV9-sh*Areg*; *n* = 7 for *Gpr174⁻/Y* mice + AAV9-sh*N* and *Gpr174⁻/Y* mice + AAV9-sh*Areg*). Scale bar, 50 μm. **d** Representative immunofluorescent images of CD31 (red), TUNEL (green), and DAPI (blue) staining and quantification of CD31⁺TUNEL⁺ cells in muscle cross sections of WT and *Gpr174⁻/Y* mice injection with AAV9-sh*Areg* 14 days after HLI (*n* = 6 for WT mice + AAV9-sh*N* and WT mice + AAV9-sh*Areg*; *n* = 7 for *Gpr174⁻/Y* mice + AAV9-sh*N* and *Gpr174⁻/Y* mice + AAV9-sh*Areg*). Scale bar, 50 μm. **e** Representative images of capillary-like structures and quantification of branch points and total tube length in mouse aortic endothelial cells stimulated with recombinant AREG for 16 h (*n* = 6). Scale bar, 200 μm. For all statistical plots, the data are presented as mean ± SD. One-way ANOVA with Bonferroni multiple comparisons test in (**a**, **b**). Two-way ANOVA with Bonferroni multiple comparisons test in (**c**, **d**). Two-tailed unpaired t-tests in (**e**). Source data are provided as a Source Data file.

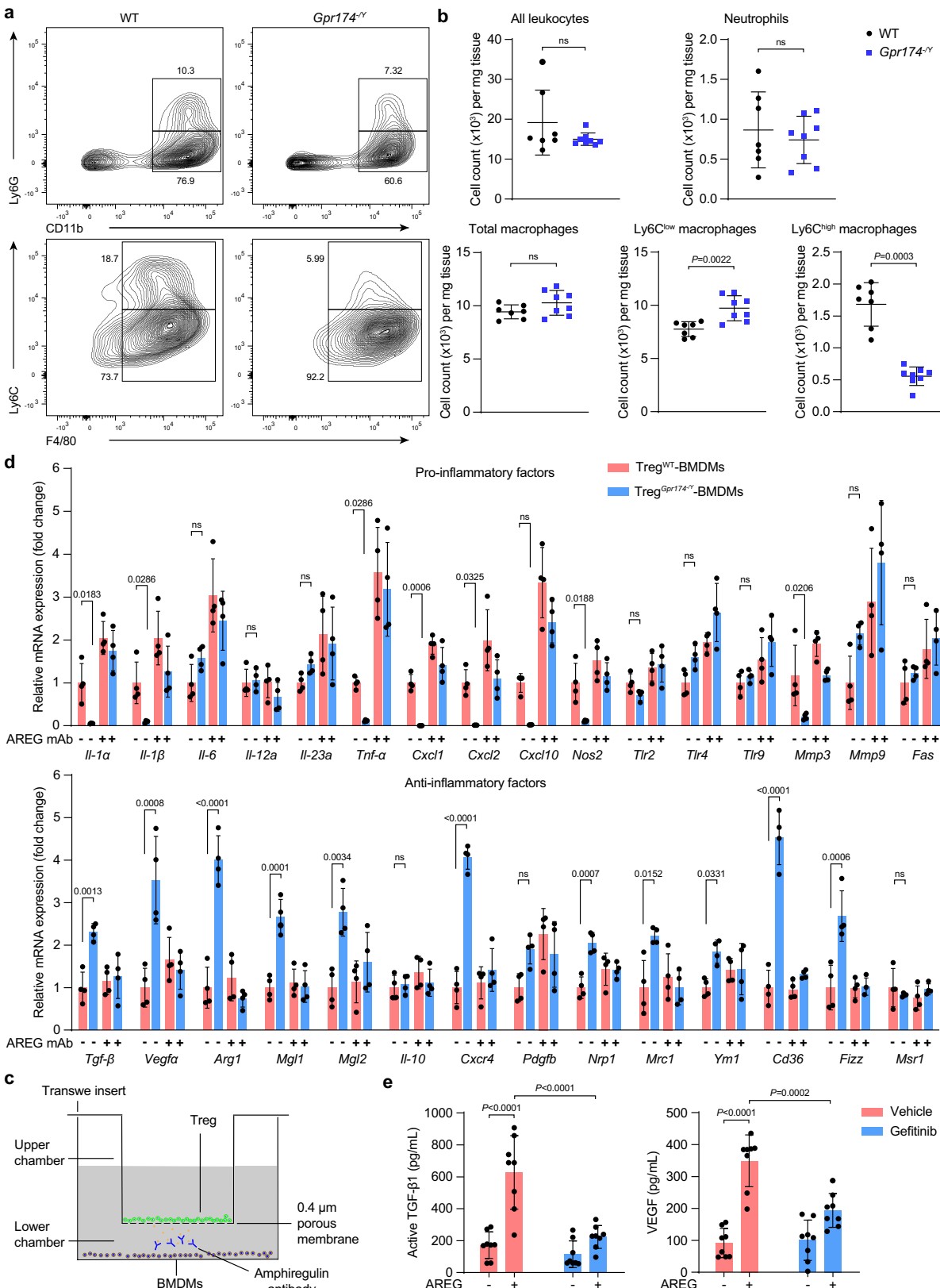

**Fig. 5 | GPR174 deficiency mitigates the inflammation in ischemic muscles through AREG. a** Representative flow cytometric dot plots of leukocyte populations in gastrocnemius tissues of WT and *Gpr174⁻/Y* mice 7 days after HLI. **b** Quantification of all leukocytes, neutrophils, total macrophages, Ly6C^low macrophages, and Ly6C^high macrophages in WT and GPR174-deficient gastrocnemius 7 days post HLI (*n* = 7 for WT mice; *n* = 8 for *Gpr174⁻/Y* mice). **c** Scheme of Tregs-macrophages co-culture. **d** Relative mRNA levels of proinflammatory (upper panel)

and anti-inflammatory (lower panel) genes in macrophages co-cultured with Tregs and AREG neutralizing antibodies for 24 h (*n* = 4). **e** Quantification of bioactive TGF-β and VEGF in macrophages stimulated with recombinant AREG in the presence or absence of inhibitors for the EGFR (Gefitinib) for 24 h (*n* = 8). For all statistical plots, the data are presented as mean ± SD. Two-tailed unpaired t-tests in (**b**). Two-way ANOVA with Bonferroni multiple comparisons test in (**d**, **e**). Source data are provided as a Source Data file.

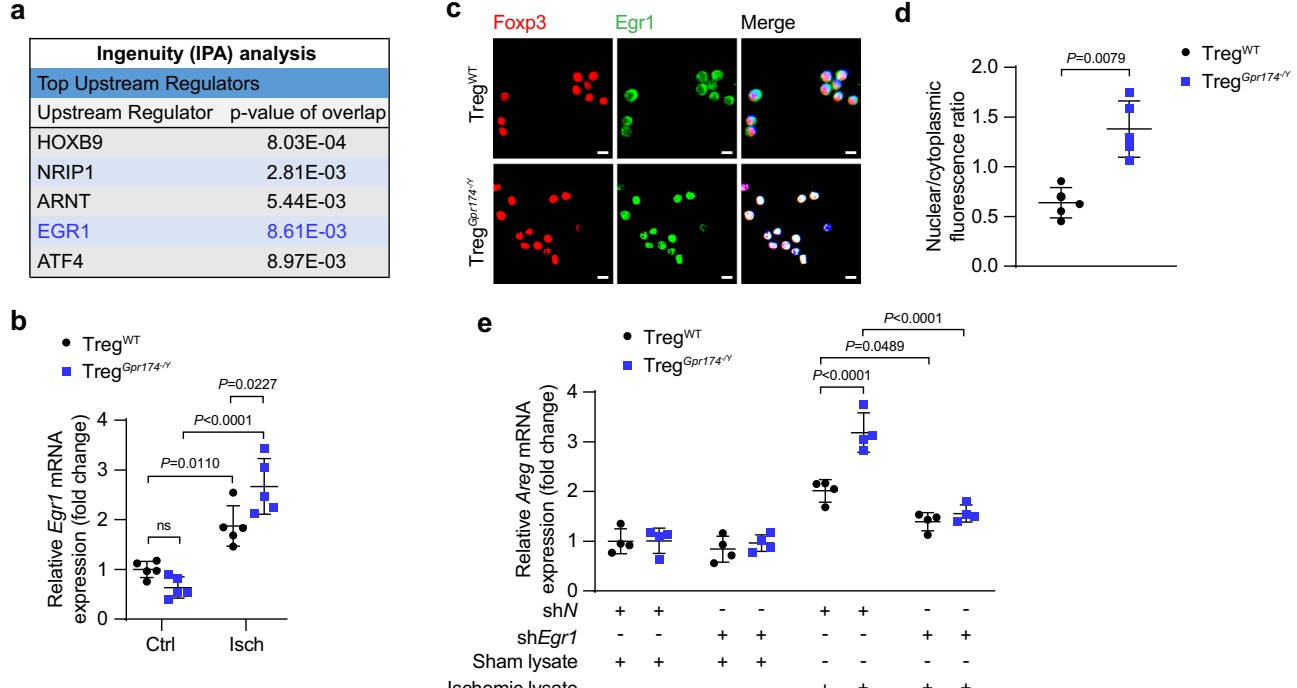

**Fig. 6 | GPR174 downregulates AREG expression by inhibiting EGR1 nuclear localization. a** Top transcription factors of AREG. **b** *Egr1* mRNA expression in Tregs isolated from the muscle tissues of WT and *Gpr174$^{-/Y}$* mice 7 days after HLI ($n = 5$). **c** Representative immunofluorescent images of EGR1 (red), and DAPI (blue) staining in Tregs isolated from the ischemic muscles of WT and *Gpr174$^{-/Y}$* mice 7 days after HLI. Scale bar, 10 μm. **d** Quantification of nuclear localization of EGR1 ($n = 5$). **e** *Areg*

mRNA expression in WT or GPR174-deficient Tregs treated with sh*Egr1* and non-ischemic or ischemic muscle lysates ($n = 4$). For all statistical plots, the data are presented as mean ± SD. Two-tailed unpaired t-tests in (**d**). Two-way ANOVA with Bonferroni multiple comparisons test in (**b**, **e**). Source data are provided as a Source Data file.

Fig. 13f, g). Collectively, GPR174 deficiency improves blood flow recovery in diabetic mice after HLI.

## Discussion

In the present study, we reveal that GPR174 negatively regulates neo-vascularization by inhibiting Treg function. We demonstrate that GPR174 deficiency improves blood flow recovery and mitigates the inflammatory response in ischemic tissues. Furthermore, we show that GPR174 limits angiogenesis and vascular remodeling and promotes inflammation by inhibiting AREG expression. Further mechanistic investigations demonstrate that GPR174 increases the activation of the Gαs/cAMP/PKA signaling pathway to inhibit the binding of EGR1 to the *Areg* promoter. Altogether, these data provide novel evidence that GPR174 inhibits neovascularization by downregulating AREG expression via the Gαs/cAMP/PKA/EGR1 signaling pathway.

Accumulating investigations have demonstrated that, in addition to their immunosuppressive effect, Tregs contribute to tissue homeostasis and regeneration by secreting growth factors[7–11]. Also, Treg infiltration has been shown to be increased in ischemic tissues, which in turn mitigates the pro-inflammatory response and enhances angiogenesis[11,13,32]. GPR174 is a GPCR that is predominantly expressed in Tregs and other lymphocyte populations[16]. Other studies have revealed that GPR174-deficient Tregs are associated with the reduced severity of autoimmune encephalomyelitis and sepsis-induced lung injury, which may be attributed to the fact that GPR174 is a negative regulator of Treg proliferation and function[17,28]. These investigations have further suggested that the function of Tregs in regulating angiogenesis may be partially mediated by GPR174.

Here, we observed that the *Gpr174$^{-/Y}$* mice exhibited reduced ischemic damage after HLI, which was attributed to improved blood flow recovery, enhanced vascular remodeling and angiogenesis, and mitigated endothelial cell apoptosis. Using adaptive transfer

experiment, we also showed that GPR174-deficient Tregs exerted protective effects in muscle tissues after ischemia. GPR174 has been identified as a Gαs-dependent receptor that elevates cAMP levels and increases PKA activity, thereby inhibiting naïve T cell activation[18]. Additionally, investigations have revealed that the cAMP/PKA signaling pathway inhibits the nuclear translocation and phosphorylation of EGR1[31,33,34]. However, whether GPR174 regulates EGR1 nuclear localization in Tregs remains unknown.

We determined the relationship between GPR174 and EGR1 using RT-qPCR and immunofluorescence analyses. GPR174-dificient Tregs isolated from the ischemic muscles exhibited increased EGR1 expression and nuclear accumulation. Further investigation of the underlying mechanism uncovered that GPR174/Gαs signaling increased cAMP levels and PKA activity to negatively regulate EGR1 nuclear accumulation in Tregs. This contributed to AREG downregulation in these cells, leading to reduced angiogenesis and delayed blood flow recovery after HLI.

An increasing number of investigations have revealed that AREG not only mitigates inflammation by inducing macrophage polarization in the lungs[21], but it also alleviates tissue damage by promoting the proliferation and differentiation of tissue precursor cells in the skin, muscles, and retinas[7,35]. AREG has also been shown to inhibit apoptosis in stem cells and liver cells[23,36]. Recent studies have shown that AREG is highly produced by Tregs in injured tissues[7,10,11]. These properties of AREG indicate that it is a key molecule involved in the regulation of Tregs to attenuate the inflammatory response and promote tissue repair. We demonstrated that AREG expression was increased in GPR174-deficient Tregs isolated from ischemic muscle compared with wild-type controls. We also found that the protective effects of the GPR174-deficient Tregs were blocked by AREG knockdown in aortic ring assay. In contrast, treatment with recombinant AREG enhanced adherens junctions and tube formation and also mitigated apoptosis in

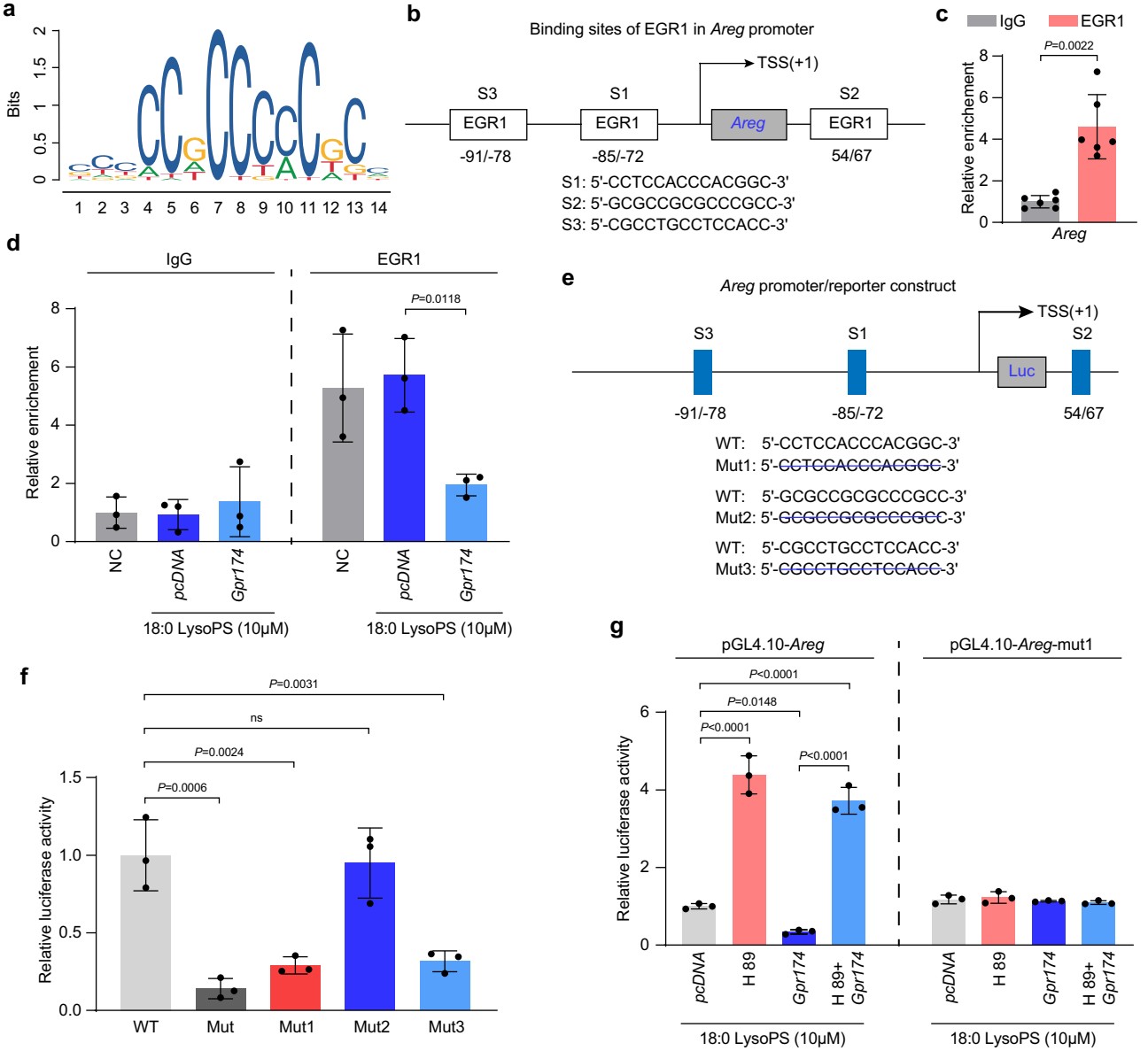

**Fig. 7 | GPR174 inhibits AREG transcription activation via PKA/EGR1 pathway.**
**a** Consensus DNA-binding motifs of EGR1 according to JASPAR database. **b** The predicted binding sites of EGR1 in *Areg* promoter according to JASPAR database. **c** ChIP-qPCR assay in HEK293A cells for EGR1 or IgG occupancy at *Areg* promoter fragments ($n = 6$). **d** ChIP-qPCR assay in HEK293A cells transfected with *Gpr174* CRISPR Activation Plasmid for EGR1 or IgG occupancy at *Areg* promoter fragments ($n = 3$). **e** Generation of luciferase reporters governed by *Areg* promoter with wild-type or mutant EGR1 binding sites; **f** Dual luciferase reporter assay in HEK293A cells

co-transduced with luciferase reporter driven by wild-type or mutated promoter, pRL-CMV-renilla, and expression plasmid for EGR1 ($n = 3$). **g** AREG promoter activity in HEK293A cells co-transfected with wild-type or mutant 1 *Areg* promoter constructs, pRL-CMV-renilla, and Gpr174 CRISPR Activation Plasmid or treated with H 89 2HCL ($n = 3$). For all statistical plots, the data are presented as mean ± SD. Two-tailed unpaired t-tests in (**c**). Two-way ANOVA with Bonferroni multiple comparisons test in (**d**, **f**, **g**). Source data are provided as a Source Data file.

endothelial cells. These findings indicated that Treg-specific GPR174 knockout directly enhanced angiogenesis by increasing AREG expression.

Our results showed that compared with wild-type Tregs, GPR174-deficient Tregs promoted anti-inflammatory polarization of macrophages after HLI, which was in line with previous finding by Dongze Qiu and colleagues in that GPR174 KO mice from the same mouse strain attenuated LPS induced inflammation response in macrophage[28]. In addition, they also showed that GPR174-deficient Tregs secreted IL-10, leading to increased M2 polarization of macrophages, thereby protecting the mice against sepsis induced lung injury[28]. In our study, IL-10 was similar between wild-type and GPR174-deficient Tregs after hindlimb ischemia, the difference might possibly be related to the degree

of inflammation response, in that inflammation response is overwhelming in sepsis model, while ischemia is dominant in our HLI model. We further demonstrated that the GPR174-deficient Tregs promoted anti-inflammatory macrophage polarization in response to ischemia via AREG. In contrast, the effects were inhibited by the EGFR inhibitor, Gefitinib, as well as an AREG neutralizing antibody. Consistent with these findings, another study reported that elevated AREG expression in NOTCH4-knockout Tregs promoted the polarization of M2 macrophages to ameliorate lung injury induced by viral infections[21].

Note that we verify the direct role of AREG in promoting endothelial cell functions. However, epidermal growth factor receptor (EGFR), an affinity receptor of AREG, was predominantly expressed in

tumor-derived endothelial cells rather than normal[37]. Thus, the underlying mechanism by which AREG acts on endothelial cells remains to be further investigated. In additional, it is to note that in our study, we only studied the impact of GPR174 knockout in male mice undergoing HLI, to minimize the influence from sex hormone variations from female mice, caution is thus needed in the interpretation of our results, and future studies are needed to explore the impact of GPR174 knockout in female mice in identical experimental conditions. Furthermore, the mouse model of hindlimb ischemia is an acute event that does not completely simulate lower extremity PAD. Therefore, clinical patient data need to be integrated into further studies to validate our results.

In conclusion, our findings demonstrate a negative regulatory role of GPR174 in response to HLI. Treg-expressed GPR174 inhibits EGR1 nuclear accumulation by coupling with Gαs to elevate cAMP levels and PKA activity, leading to the downregulation AREG expression and thereby reduced neovascularization post ischemic insult. These data suggest that targeting GPR174 may be a potential therapeutic strategy for treating ischemic vascular diseases, such as PAD.

## Methods

### Experimental animals

All animal experiments were approved by the Ethics Committee of Zhongshan Hospital, Fudan University, Shanghai, China (2020-045) and were following the guidance for the Care and Use of Laboratory Animals published by the National Research Council (U.S.) Institute for Laboratory Animal Research. Mice were housed with free access to food and water, under an alternating 12 h light−dark cycle in a constant temperature ($22 \pm 1\,^{\circ}C$) and 50% relative humidity room. $Gpr174^{-/Y}$ mice with C57BL/6 genetic background were acquired from Zhenju Song Laboratory, Zhongshan Hospital, Fudan University (Supplementary Fig. 1a). Male C57BL/6 $Foxp3^{GFP}$ reporter mice aged 8 to 10 weeks were used to isolate $CD4^{+}Foxp3^{GFP}$ Tregs by flow cytometry (Supplementary Fig. 1b). Male C57BL/6 wild type and $Gpr174^{-/Y}$ mice aged 8 to 10 weeks were used for sorting Tregs using EasySep™ Mouse $CD4^{+}CD25^{+}$ Regulatory T Cell Isolation Kit II (STEMCELL Technologies, 18783). Eight-week-old male C57BL/6 $Rag1^{-/-}$ mice obtained from Cyagen Biosciences were injected with Tregs to exam the effects of GPR174. Male C57BL/6 wild-type and $Gpr174^{-/Y}$ mice aged 4 weeks were intravenously injected with adeno-associated viral (AAV9) bearing shRNA against AREG[38] (the shRNA sequence was 5′-CCACAAATATCCGGCTATATT-3′) to downregulate AREG expression in muscle tissues. All experiment animals were male for the character of less influence of sex hormone[39,40].

### Hindlimb ischemia

To generate a model of hindlimb ischemia, we ligated the left hindlimb femoral artery of mice. Briefly, male C57BL/6 wild-type and $Gpr174^{-/Y}$ mice aged 8 to 10 weeks were anesthetized with isoflurane and placed on the heated blanket to maintain body temperature; after this, the hind limb was shaved, and, following a small incision (≈1 cm) was made in the skin to directly expose the femoral vasculature. Both the Proximal and distal sites of the femoral artery were ligated using surgical knots, followed by dissociating and excising the segment between the ligation sites and suturing the skin with noncontinuous absorbable suture. Hindlimb blood flow was assessed by laser Doppler perfusion imaging (PeriScan PIM 3, Perimed) at 0, 1, 3, 7, 14 days post HLI. The rate of reperfusion in the hind limb was measured by analyzing the ratio of blood flow in the ligated versus non-ligated limb using PIMSoft (Perimed). At day 3, 7, and 14 post HLI, mice undergoing surgery were sacrificed by cervical dislocation, and the gastrocnemius and adductor muscles from both legs were harvested for subsequent analysis.

### Induction of diabetic mouse model

To induce type I diabetes (T1DM) mouse model, we used a standard protocol of streptozotocin (STZ) (Sigma, S0130) injection. Briefly,

six-week-old male C57BL6 $Gpr174^{-/Y}$ mice and WT littermates were fasted for 6 h and then injected intraperitoneally (i.p.) with citrate buffer or STZ (55 mg/kg in sodium citrate buffer, pH 4.5) for 5 consecutive days. Two weeks after the first injection non-fasting blood glucose concentrations were measured by an electronic glucometer (Roche Diabetes Care GmbH, Mannheim, Germany), and only mice with blood glucose levels > 280 mg/dL were considered diabetic and used for experiments. Surgical intervention to induce unilateral hindlimb ischemia was performed 12 weeks after constant hyperglycemia.

### Cells culture

Human umbilical vein endothelial cells (HUVECs) (Cyagen Biosciences, HUVEC-20001) were obtained from Cyagen Biosciences and cultured in endothelial cell medium (ECM, Sciencell, 1001) supplemented with 5% fetal bovine serum (FBS, Sciencell, 0025), 1% endothelial cell growth supplement (ECGS, Sciencell, 1052) and 1% antibiotic solution (P/S, Sciencell, 0503). Next, HUVECs were stimulated with 50 ng/mL AREG (Peprotech, 100-55B) to evaluate apoptosis and adherent junctions between endothelial cells. Tregs were isolated from ischemic muscle or the spleens of male C57BL/6 wild type and $Gpr174^{-/Y}$ mice aged 8 to 10 weeks using EasySep™ Mouse $CD4^{+}CD25^{+}$ Regulatory T Cell Isolation Kit II (STEMCELL Technologies, 18783). The cells were stimulated with Dynabeads Mouse T-Activator CD3/CD28 (bead-to-cell ratio of 2:1, Gibco, 11453D) for 4 days in the presence of 40 ng/mL IL2 (Peprotech, 212-12). Cells were cultured in Advanced medium RPMI-1640 (Gibco, 12633012) supplemented with 10% FBS, 2-mercaptoethanol, l-glutamine, and penicillin/streptomycin.

### Single cell suspensions preparation

Single cell suspensions from spleen or ischemic muscle were prepared as follows. For muscle, mice were culled by cervical dislocation, and the injured gastrocnemius muscle was immediately excised and was minced in a 2 mL tube on ice. Next, the minced muscle was enzymatically digested in PBS buffer containing Dispase II (Sigma, D4693) and Collagenase IV (ThermoFisher Scientific, 17104019) for 40 min at 37 °C with gentle shaking (120 rpm) on shaker (SHANGHAI MINQUAN INSTRUMENT CO., LTD, MQD-S2R). After digestion, an equal volume of 20% FBS in HBSS was used to stopped the reaction and the suspension was passed through a 70 µm cell strainer (Corning Falcon, 352350) and 40 µm cell strainer (Corning Falcon, 352340). For spleen, after mice were sacrificed, the connective and adipose tissues of the spleen were removed. Then the spleen was transferred to a 100 mm culture dish containing 10 mL cold PBS and was cut into small pieces. The spleen tissue was ground with a 2.0 mL syringe plunger and aggregates and debris were removed by passing cell suspension through a 70 µm cell strainer and 40 µm cell strainer. Cell suspensions were collected by centrifugation at × 100 g for 5 min at 4 °C (Thermo Fisher Scientific, 75004250) and resuspended on ice with 10 mL 1 × lysing buffer (BD Biosciences, 555899) for 5 min to remove erythrocytes. After centrifugation at × 100 g for 5 min at 4 °C, the filtered cell suspension was resuspended in stain buffer (PBS) (BD Biosciences, 554656) and transferred to a 1.5 mL EP tube. The obtained single cell suspensions were used for further analysis and sorting with flow cytometry.

### Flow cytometry

Single cell suspensions of spleen and muscle were prepared as above (see Single cell suspensions preparation section). Prior to surface marker staining with antibodies, CD16/CD32 Monoclonal Antibody (FRC-4G8) (1:50 dilution, Thermo Fisher Scientific, MFCR00-4) was used to block Fc gamma receptors at 4 °C for 15 min. Then, single cell suspensions were incubated with live/dead-BV510 (1:1000 dilution, BD Biosciences, 564406) at room temperature for 10 min. Thereafter, cells were incubated with the appropriate fluorophore-conjugated antibodies at 4 °C for 30 min. Flow cytometric analysis was performed on

FACSAria™ III flow cytometer (BD Biosciences) and cells sorted using a moflo Astrios EQ (Beckman Instruments, Inc). Data were analyzed with FlowJo 10 software (Tree Star). All fluorophore-conjugated antibodies are listed in Supplementary Table 1.

## Isolation of mouse regulatory T cells
Mouse regulatory T cells were sorted using EasySep™ Mouse CD4+CD25+ Regulatory T Cell Isolation Kit II (STEMCELL Technologies, 18783) according to manufacturer's instructions. Single cell suspension (1 mL) was prepared as above and resuspended with rat serum (50 μL/mL). And CD4+ T Cell Isolation Cocktail (50 μL/mL of sample) was added and incubated at room temperature (RT) for 10 min. Then, Streptavidin RapidSpheres™ (75 μL/mL of sample) was added and incubated at RT for 2.5 min. EasySep™ Buffer (STEMCELL Technologies, 20144) was added to top up sample to 2.5 mL and the tube was placed into the EasyEights™ EasySep™ Magnet (STEMCELL Technologies, 18103) and incubated at RT for 5 min. Thereafter, the enriched cell suspension was carefully transferred into a new 5 mL tube and was collected by centrifugation at × 200 g at RT for 10 min (Thermo Fisher Scientific, 75004250) and was resuspend with 0.5 mL EasySep™ Buffer. Cell suspension was incubated with 25 μL FcR blocker to block Fc gamma receptors at RT for 5 min. Then, CD25 Regulatory T Cell Positive Selection Cocktail (25 μL of sample) was added and incubated at RT for 10 min. PE Selection Cocktail (10 μL of sample) and Dextran RapidSpheres™ (30 μL of sample) were sequentially added and incubated at RT for 5 min. Cell suspension was then placed into the magnet and incubated at RT for 15 min. Finally, the cells were washed three times with 2.5 mL EasySep™ Buffer and supernatant was carefully discarded.

## Adaptive transfer regulatory T cells
For Tregs transfer experiments, Tregs were freshly isolated from spleens of 8- to 10-week-old male C57BL/6 wild type and Gpr174−/Y mice using EasySep™ Mouse CD4+CD25+ Regulatory T Cell Isolation Kit II (STEMCELL Technologies, 18783) as above. Then, 1 × 10⁶ Tregs diluted in 50 μL sterile PBS were injected into 8-week-old male C57BL/6 Rag1−/− mice calf muscles 3 days after HLI.

## Isolation of mouse aortic endothelial cells
Mouse aortic endothelial cells were isolated from the aorta of 8-week-old male C57BL/6 wild type mice. In brief, mice were anesthetized and Heparin (100 U/mL in sterile PBS) was injected into the left ventricle to remove circulating blood. The aortae were dissected along the longitudinal axis in DMEM and incubated with collagenase type II (2 mg/mL, Worthington, LS004176) for 45 min at 37 °C. Detached endothelial cells were collected centrifugation at × 112 g (Thermo Fisher Scientific, 75004250) for 5 min, resuspended in 20% FBS-DMEM, and then cultured in endothelial cell medium (ECM, Sciencell, 1001) supplemented with 5% fetal bovine serum (FBS, Sciencell, 0025), 1% endothelial cell growth supplement (ECGS, Sciencell, 1052) and 1% antibiotic solution (P/S, Sciencell, 0503) for 1 week.

## Aortic sprouting assay
Generation of thoracic aortas segments and aortic sprouting assay were performed as follows. In brief, male C57BL/6 wild type mice aged 8 to 10 weeks were killed by cervical dislocation, and thoracic aortas were flushed with Heparin (100 U/mL in sterile PBS) to remove all blood and were removed all extraneous tissue. Segments of ≈0.5 mm were cut and embedded in reduced growth factor Matrigel (Corning Falcon, 356231) in lower chamber and incubated for 1 h at 37 °C. Subsequently, endothelial cell medium (ECM, Sciencell, 1001) containing 5% fetal bovine serum (FBS, Sciencell, 0025), 1% endothelial cell growth supplement (ECGS, Sciencell, 1052) and 1% antibiotic solution (P/S, Sciencell, 0503) was added to the lower chamber and 1 × 10⁴ Tregs isolated from ischemic muscle of 8- to 10-week-old male C57BL/6 wild

type and Gpr174−/Y mice using EasySep™ Mouse CD4+CD25+ Regulatory T Cell Isolation Kit II (STEMCELL Technologies, 18783) were cultured in the upper chamber containing Dynabeads Mouse T-Activator CD3/CD28 (bead-to-cell ratio of 2:1, Gibco, 11453D), and 40 ng/mL IL-2 (Peprotech, 212-12). Images of sprouting aortas were obtained after 5 days.

## Tubule formation assay
The mixture of a total of 1 × 10⁴ HUVECs or mouse aortic endothelial cells (MAECs) and 50 μL endothelial cell medium (Sciencell, 1001) containing 50 ng/mL AREG (Peprotech, 100-55B) were plated into each well of μ-Slide Angiogenesis (ibidi, 81506) with 10 μL Matrigel (Corning Falcon, 356231). Images were captured after 16 h of culture and were analyzed by Image J software version 1.53 (National Institutes of Health, USA) for the quantification of tube networks and total tubule length.

## Ex-vivo Matrigel plug assay
Matrigel plug neovascularization was performed as follows. Briefly, eight- to ten-week-old male C57BL/6 wild-type and Gpr174−/Y mice were anesthetized with isoflurane and were injected subcutaneously with 200 μL reduced growth factor Matrigel (Corning Falcon, 356231) containing 30 ng/mL VEGF (Pre-protech, 450–32) and 50 U heparin (Pfizer), which quickly formed a solid plug at body temperature. Matrigel plugs were extracted after 14 days and homogenized in 0.5 mL of cell lysis buffer and centrifuged at × 6000 g at 4 °C for 60 min. Hemoglobin was detected at 400 nm wavelength by using hemoglobin assay kit (Sigma-Aldrich, MAK115). Histological analysis of Matrigel plugs was performed using hematoxylin and eosin.

## Isolation of bone marrow-derived macrophages (BMDMs)
Bone-marrow-derived macrophages (BMDMs) were isolated from the femur and tibia of 8-week-old male C57BL/6 wild type mice. In brief, bone marrow cells were flushed out of the femur and tibiae bones of WT mice with sterile PBS and cultured in RPMI 1640 medium supplemented with 10% FBS, 100 Units/mL penicillin/streptomycin, and 10 ng/mL macrophage colony stimulating factor (M-CSF) (Sigma, SRP3221) for 3 days to allow differentiation into macrophages.

## In vitro macrophages coculture and treatment
Macrophages derived from BMDMs were stimulated with 50 ng/mL AREG (Peprotech, 100-55B) and Gefitinib (Selleck, S1025) (1 mM). After 24 h of treatment, the conditional medium was collected for ELISA. Macrophages were used for co-culture with Tregs from indicated mice in a ratio of macrophage:Treg at 2:1 in the presence or absence of neutralizing antibodies against AREG (R&D Systems, AF989) (2 μg/mL) to detect anti- and pro-inflammatory factors expression. After 24 h of coculture and treatment, cells were collected for RT-qPCR analysis.

## Immunofluorescence staining
Tregs sorted from the non-ischemic or ischemic muscles using EasySep™ Mouse CD4+CD25+ Regulatory T Cell Isolation Kit II (STEMCELL Technologies, 18783). Cells were centrifuged at × 88 g for 5 min, in a Cytospin centrifuge (Thermo Shandon Cytospin 3) onto gelatin-coated Cytospin slides. The slides were air-dried and fixed in 4% paraformaldehyde for 10 min at room temperature. Thereafter, fixed cells were permeabilized with 0.1% Triton X-100 in PBS for 10 min and blocked in 1% BSA for 1 h at room temperature. Then, cells were stained with Primary antibodies at overnight 4 °C. Subsequently, cells were incubated with the appropriate fluorescent secondary antibody for 1 h at room temperature.

Muscle tissues were quickly harvested and embedded in optimum cutting temperature (OCT) compound (Sakura, Torrance, CA, USA, 4583). Then, immunofluorescent staining was performed on 7 μm mouse muscle cryosections. Sections were blocked with 1% BSA for 60 min, permeabilized with 0.1% Triton X-100 in PBS for 15 min and

then incubated with primary antibodies overnight at 4 °C. Later, the tissue sections were incubated with the appropriate fluorescent secondary antibodies (Invitrogen, Carlsbad, CA, USA) for 1 h at room temperature and avoid exposure to light. Subsequently, DAPI was used to label the nuclei.

Muscle hypoxia was detected using HypoxyprobeTM Plus Kit (Hypoxyprobe, Inc, HP2-100Kit) according to the manufacturer's protocol. Briefly, mice received an intraperitoneal injection of 100 mg/kg pimonidazole HCl (20 mg/mL in 0.9% NaCl). One hour after injection, gastrocnemius tissues were excised and embedded as described above. Hypoxyprobe was detected by incubating the samples with anti-pimonidazole FITC-conjugated mouse IgG1 monoclonal antibody (FITC-Mab1) and rabbit anti-FITC conjugated with horseradish peroxidase. TUNEL staining was performed with One Step TUNEL Apoptosis Assay Kit (Beyotime, C1088) according to the manufacturer's instructions. Immunofluorescent images were captured using a laser scanning confocal microscope (Germany, Zeiss LSM710). The primary antibodies used for these experiments are listed in Supplementary Table 2.

## Western blotting
Cells lysed on ice with RIPA lysis buffer containing PMSF, protease inhibitors, and phosphatase inhibitors and the muscle tissues were grinded in a homogenizer (Servicebio, KZ-5F-3D) according to the manufacturer's instructions with RIPA lysis buffer. Lysates were centrifuged for × 14,000 g (Thermo Fisher Scientific, 75004250), 15 min at 4 °C and the supernatant was collected for Western blotting. Protein concentration was determined by pierce BCA assay. Proteins were separated by SDS-PAGE and transferred to polyvinyl difluoride membranes (PVDF) (Millipore, 3010040001). Targeted proteins were specifically incubated with the Primary antibodies overnight at 4 °C and the corresponding secondary antibodies for 1 h at room temperature. Then, images of chemiluminescent western blots were acquired from the ChemiDoc XRS + System (BIO-RAD) which is controlled by Image Lab Software (BIO-RAD). Protein levels were quantified using Image J software version 1.53 (National Institutes of Health, USA). The primary antibodies are listed in Supplementary Table 3 and the uncropped scans of all blots are provided in the Source Data file.

## RNA extraction and quantitative RT-PCR
RNA of cultured macrophages was isolated using RNeasy Mini Kit (Qiagen, 74104). Total RNA of Tregs from muscle of indicated mice was extracted using RNeasy Plus Micro Kit (Qiagen, 74034). RNA from muscle tissues was isolated using Trizol reagent (Ambion, 15596018). RNA was reverse-transcribed to cDNA by Reverse Transcription Kit (Takara, RR047A). Quantification RT-qPCR analysis was performed using Bio-Rad's CFX96 (Bio-Rad) by mixing SYBR Green master mix (Yeasen, 11201ES03), equal amount of cDNAs, and primers. To compensate for variations in RNA input and efficiency of reverse-transcription, 18S was used to normalized the mRNA expression level of genes. The results were calculated as relative expression or fold change using the delta-delta CT method. The sequences of primer are provided in Supplementary Table 4.

## Enzyme-linked immunosorbent assay (ELISA)
The serum of 8- to 10- week-old male C57BL/6 wild-type mice was collected 5 days after HLI and the levels of CCL21 were determined using mouse CCL21/Exodus-2 ELISA Kit (ABclonal, RK00166), according to the manufacturer's instructions. The levels of VEGF and bioactive TGF-β1 in the medium of macrophages or in serum of mice were detected using Mouse VEGF Quantikine ELISA Kit (R&D Systems, MMV00) and Mouse TGF-beta 1 ELISA Kit (Absion, abs520021). The levels of AREG and IL-10 in serum of recipient mice 10 days after HLI were determined using Mouse amphiregulin ELISA kit (BOSTER, EK0591) and Mouse IL-10 ELISA kit (ABclonal, RK00016). Following the addition of stop solution, the absorbance was measured at 450 and

570 nm using FlexStation3 (Molecular Devices) and the data was collected using SoftMax Pro 5 (Molecular Devices).

## LC-MS/MS analysis
Liquid chromatography-tandem mass spectrometry (LC-MS/MS) analysis was performed and analyzed by Suzhou PANOMIX Biomedical Tech Co., LTD. Each muscle sample (100 mg) obtained from 8- to 10-week-old male C57BL/6 wild-type mice 7 days after HLI and control mice was transferred into 2 mL centrifuge tubes, add 750 μL of Chloroform methanol mixed solution (2:1) (pre-cooled at −20℃), vortex for 30 s and put on the ice for 40 min, add 190 μL $H_2O$, and still put on the ice for 10 min. We centrifuged Lysates at × 13778 g for 5 min at room temperature and transfer 300 μL lower layer fluid into a new centrifuge tube, and then Add 500 μL of Chloroform methanol mixed solution (2:1) (pre-cooled at −20℃), vortex for 30 s. After centrifuged at × 13778 g for 5 min at room temperature, 400 μL lower layer fluid was transferred into the same centrifuge tube above. Samples were concentrated to dry in vacuum and were dissolved with 200 μL isopropanol. The supernatant was filtered through 0.22 μm membrane to obtain the prepared samples for LC-MS. The samples were loaded onto analytical column (2.1 × 100 mm, ACQUITY UPLC® BEH C18 column, 1.7 μm, Waters) at a flow rate of 0.25 mL/min under a linear gradient (0–5 min, 70–57% C; 5–5.1 min, 57–50% C; 5.1–14 min, 50–30% C; 14–14.1 min, 30% C; 14.1–21 min, 30–1% C; 21–24 min, 1% C; 24–24.1 min, 1–70% C; 24.1–28 min, 70% C). The lipid analysis was conducted on an Orbitrap Fusion Lumos mass spectrometer (Thermo Finnigan, San Jose, CA) in data-dependent acquisition system. An MS survey scan was obtained for the m/z range 150–2000 at a mass resolution of 35,000.

## RNA sequencing analysis
RNA sequencing was performed and analyzed by BGI Tech Inc. Briefly, total RNA was isolated from ischemic gastrocnemius samples of 8- to 10- week-old male C57BL/6 Gpr174−/Y and WT mice 7 days after HLI ($n = 3$ in each group). RNA-sequencing was performed on BGISEQ-500 platform. DEGseq was used to identify differentially expressed genes, with $q$-value<0.05 and $|log_2Fc| > 1$.

## Cell transfection
Isolation and culture of mouse Tregs were performed as described above. Control Lentiviruses-shRNA (Santa Cruz, sc-108080) and Lentiviruses-shEgr1 (Santa Cruz, sc-35267-V) were transfected into Tregs and RT-qPCR was performed to determine lentivirus-mediated Egr1 knockdown in Tregs (Supplementary Fig. 10). Lipofectamine™ 3000 Transfection Reagent (ThermoFisher Scientific, L3000008) was used for transfection of plasmids into HEK293A cells (Shanghai Zhong Qiao Xin Zhou Biotechnology Co., Ltd., ZQ0941). For chromatin immunoprecipitation (CHIP), control CRISPR activation plasmid (Santa Cruz, sc-437275), and Gpr174 CRISPR activation plasmid (Santa Cruz, sc-413678-ACT) were transfected into HEK293A cells for 24 h, and then cells were treated with 10 μM LysoPS (Avanti Polar Lipids Inc., 858144) for 24 h and were collected for CHIP. For Areg promoter luciferase activity assay, pGL4.10-Areg-wt (or pGL4.10-Areg-mut, mut1, mut2, mut3), pRL-CMV, and pcDNA3.1-Egr1 designed and constructed by OBiO Technology (Shanghai) Corp.,Ltd were co-transfected into HEK293A cells (Supplementary data). Then, 24 h post transfection, cells were stimulated with 10 μM LysoPS for 24 h and collected for measurement of luciferase activity.

## Chromatin immunoprecipitation
Chromatin immunoprecipitation was performed using a CHIP assay Kit (ThermoFisher Scientific, 26156) according to the instructions provided by the manufacturer. Briefly, cells were incubated with 1% formaldehyde (ThermoFisher Scientific, 28908) for 10 min at room temperature to cross-link proteins to DNA. The cross-linked chromatin preparation was digested with optimal micrococcal nuclease

concentration to generate DNA fragments of 200–1000 bp (Supplementary Fig. 12a). Optimal digestion for chromatin fragmentation was determined by detecting different concentration of micrococcal nuclease and the one that generated DNA fragments from 200 to 1000 bp with a more intense ladder of bands occurring at approximately 160, 320, and 480 bp was considered optimal (Supplementary Fig. 12a). The digested cross-linked chromatin was immunoprecipitated with target specific anti-EGR1 antibody (1:50 dilution, Cell Signaling Technology, 4154 s), 10 μL Anti-RNA Polymerase II Antibody (CHIP assay Kit content, ThermoFisher Scientific, 26156) as positive control, and 1 μLnormal rabbit IgG antibody (CHIP assay Kit content, ThermoFisher Scientific, 26156) as negative control. The immunoprecipitated complex was then collected by using ChIP Grade Protein A/G Plus Agarose. Subsequently, the immunoprecipitated chromatin was eluted from the antibody/Protein A/G Plus Agarose followed by reversal of cross-links in 65 °C for 1.5 h and purification using DNA Clean-Up column. The purified DNA was amplified by RT-qPCR using specific primers for the promoter of *Areg*. The enrichment percentage were expressed as 10% of the total input DNA. Primers of *Areg* promoter used for ChIP-qPCR assay: 5′-TTGATACTCGGCTCAGGCCA-3′; 5′-ATCCATCAGCACTGTGGTCC-3′.

### Dual luciferase assay

Transfection for dual-luciferase reporter plasmids was performed as described above. Luciferase activity was measured using the Dual-Luciferase Reporter Assay System (Promega, E1910) according to the manufacturer's instructions. In brief, HEK293A cells were lysed by passive lysis buffer and harvested the lysate. The firefly and renilla luciferase activity in the lysate were detected using GloMax 20/20 Luminometer (Promega). Renilla luciferase activity was used as an internal control to compensate for variations in transfection efficiency. The luciferase activity of firefly was normalized to renilla luciferase activity to determine the target gene promoter activity. Luciferase activity of promoter was expressed as fold changes of control group.

### Statistical analysis

All data are presented as scatter dot plot or line chart with mean ± SD, and $p < 0.05$ was considered significant. As to comparison between two groups, two-tailed unpaired t test was used as indicated. For more than two groups, one-way ANOVA with Bonferroni multiple comparisons test was used and for experiments with a second variable, two-way ANOVA with Bonferroni multiple comparisons test was performed. For Laser Doppler imaging experiments, two-way repeated-measures ANOVA with Sidak's multiple comparisons test was used. Statistical analysis was performed with GraphPad Prism 8.0 (GraphPad Software, San Diego, CA, USA).

### Reporting summary

Further information on research design is available in the Nature Portfolio Reporting Summary linked to this article.

## Data availability

The transcriptome (RNA-seq) data generated in this study have been deposited in NCBI's Gene Expression Omnibus (GEO) under accession code GSE214684. Metabolomics data have been deposited to the EMBL-EBI MetaboLights database with the identifier MTBLS6103. Previously published dataset used in this study is accessible at the GEO under accession number: GSE50096[7]. The transcription factors that may possibly bind to *Areg* promoter were predicted using Ingenuity Pathway Analysis [http://www.ingenuity.com/]. The binding sites of EGR1 within the *Areg* promoter were predicted using The JASPAR database [https://jaspar.genereg.net/]. Source data are provided with this paper.

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

## Acknowledgements
This work was supported by grants from a key project of the National Natural Science Foundation of China (82130010, A.J.S.), the National Science Fund for Distinguished Young Scholars to A.S. (81725002, A.J.S.), Shanghai Science and Technology Commission (21MC1930400, Z.J.S.), National Natural Science Foundation of China (81900353, X.L.S.), and the Innovation Program of Shanghai Municipal Education Commission (A.J.S.).

## Author contributions
J.L., Z.J.S., and A.J.S. designed the research; J.L., L.H.P., W.X.H., and S.Q.C. performed experiments; J.L., L.H.P., W.X.H, S.Q.C., Z.J.S., and A.J.S. analyzed and discussed the results; J.L., L.H.P., and A.J.S. wrote the article; W.X.H., S.Q.C., P.Y.B., W.L., X.L.S., F.R.H., X.L.J., J.L.C., Y.J.C., K.H., Z.J.S., and J.B.G. provided technical support and contributed to the discussion of the article; L.H.P., Y.J.C., K.H., Z.J.S., and A.J.S. did critical editing.

## Competing interests
The authors declare no competing interests.
