## [Peer Review File · Nature Communications]

GPR174 Knockdown Enhances Blood Flow Recovery in Hindlimb Ischemia Mice Model by Upregulating AREG ExpressionReviewer #1 (Remarks to the Author):

The manuscript by Liu et al. seeks to demonstrate the the G-coupled protein receptor GPR174 deletion increases blood flow recovery in mice following hindlimb ischemia. They go on to describe a, probably not the only, mechanism. I offer the following suggestions and comments to the authors.

1) The problem in PAD is impaired blood flow and the findings from the hind-limb ischemia model are that genetic deletion accelerates the rate of perfusion recovery over normal. The study does not report on impaired perfusion recovery or impaired angiogenesis. This occurs because the mice are quite young at the time of the HLI induction. Also, the authors used only male mice which by itself is not a limitation but added to the prior is more of an issue. Perhaps the authors could study the mice at more advanced age; add high fat diet or streptozocin. The other reason to make this point is the same mouse strain has been shown to have improved survival with a lesser cytokine response to LPS. This is clearly linked to an M1-like macrophage response and a greater M1-like response is linked to poorer perfusion recovery; making the major finding if not mechanism predictable.

2) GPR174 belongs to a G protein-coupled receptor superfamily which raises the question as to whether the genetic deletion results in changes in expression of the related GPR (GCPR, being the better abbreviation). An examination of a few related ones would be ideal.

3) Differences in the "early" response to HLI is often linked to macrophages. Are the macrophages from genetic deletion mice similar to the wild type, in the absence of HLI?

4) A number of the studies are done on mice Day post-HLI but at that time point the perfusion recovery in knock out vs. controls is already different. This raises the possibility that the differences in perfusion recovery are not the cause of the outcome difference but a result of the outcome difference. Redoing all the studies is not necessary but selecting a few outcomes at Day 3 (like comment 1) would strengthen the overall findings.

5) The wording on the transition from the HLI studies to the Matrigel needs to be modified. Most would consider HLI the in-vivo model and the matrigel an ex-vivo model. Perhaps what they mean is to begin to elucidate the mechanism of the effects.

Reviewer #2 (Remarks to the Author):

In the current manuscript Liu et al. demonstrate that GPR174 negatively regulates neovascularization by inhibiting Treg function. They demonstrate that GPR174 deficiency significantly improves blood flow recovery and mitigates the inflammatory response in ischemic tissues. Furthermore, they show that GPR174 limits angiogenesis and vascular remodeling and promotes inflammation by inhibiting AREG expression. Mechanistic investigations demonstrate that GPR174 increases the activation of the Gas/cAMP/PKA signaling pathway to inhibit the binding of EGR1 to the Areg promoter.

As such this manuscript provides strong evidence that GPR174 inhibits neovascularization by downregulating AREG expression via the Gas/cAMP/PKA/EGR1 signaling pathway.

The manuscript is very well written, the experimental set-up and build-up of the arguments is excellent and provide a very complete set of data that support their conclusions.

I am impressed the thorough approach and completeness of the data set presented and have no further comments, except one minor point:

The authors report in the introduction : " Tregs have also been reported to enhance the regeneration of peripheral vasculature in PAD12,13", it would correct to refer also to the first two papers addressing a role for Tregs in the regulation of neovascularization :

1. Zouggari Y, Ait-Oufella H, Waeckel L, et al. Regulatory T cells modulate postischemic neovascularization. *Circulation*. 2009;120:1415–25.
2. Hellingman AA, van der Vlugt LE, Lijkwan MA, Bastiaansen AJ, Sparwasser T, Smits HH, et al. . A limited role for regulatory T cells in post-ischemic neovascularization. *J Cell Mol Med*. (2012) 16:328–36. 10.1111/j.1582-4934.2011.01300.x

Reviewer #3 (Remarks to the Author):

This is a strong report by Lui and colleagues, and I congratulate the authors for this well-performed study. The authors claim that activation of regulatory T cells via a deletion of GPR174, an inhibitory receptor, leads to Treg activation and production of Areg. This, in turn, promotes the neovascularization in mice after HLI. They showed that Areg indeed promotes endothelial cell functions, and the authors could block this convincingly with an sh-AAV vector system. In addition, they investigated the molecular footprint at the Areg gene locus and identified EGR1 as an initiator of Areg gene expression. This work is of high significance to the field of regenerative medicine and coagulation medicine, and it is overall well supported by statistical tests and advanced imaging and laboratory techniques. Although the importance of Treg in the field of ischemia and reperfusion injury is known, this study extends on the finding that Treg cells are also important after HLI. Again, the authors identify GPR174 as an inhibitory receptor, another fact already known, and the effects on myeloid populations (e.g. Qiu et al., „Gpr174-deficient regulatory T cells decrease cytokine storm in septic mice, "Nature 2019). This being said, the study is a well-performed investigation of the GPR174-coupled Treg function in HLI and is thus relevant for the field.

My remarks in details:

- In Figure 2, the authors convincingly show that Gpr174 mRNA is expressed in Treg cells upon Isch, but not Ctrl. Why do they not include other regeneration-related cell types such as ILC2s and myeloid cell populations? It would be great to add these to the figure and this would be a requirement for publication. Also, this is important to address Gpr174 in the future (as a therapeutic option). Is there a good flow cytometry antibody? That would help. Otherwise, sorting and mRNA are also good.
- All figures: please show standard deviation instead of SEM.
- All figures: please add % into the sorting gates, where shown (missing in Figure S2, for example)
- Line 127: please relabel „Figure 2L-N" into „Figure 1L-N"
- Line 135, 145: what are Foxp3+ mice? Do you mean Foxp3(GFP) reporter animals? Please verify this, add the citation to identify these mice, and be cautious throughout the ms when you refer to sorted Foxp3+ Treg cells: if you sorted Foxp3(GFP)+ Treg cells, this makes sense; if you sorted Foxp3+ Treg cells, this would require fixation and permeabilization and sorting for RNA would be useless
- I do not understand Figure 5E: what is the difference between the left and right groups in each plot? Did you forget to add another discriminator (e.g. the GPR174 KO vs WT?)
- Figure 3A: I, personally, would also show the full RNA-seq results from the supplement and then highlight Areg and Egr1. Just a recommendation.

Reviewer #4 (Remarks to the Author):

Liu et al investigate in this study whether GPR174 affects ischemic disease by regulating

Treg function. The authors provide evidence that GPR174 deficiency in Tregs results in reduced inflammation and improved endothelial function and suggest that this is due to a signalling cascade involving GPR174/PKA-dependent EGR1 translocation with consecutive upregulation of AREG expression.

This is a very thorough and well-written study, the topic is interesting and relevant, the methods are state-of-the art. I have only a few major points and some minor issues:

Major points:

Fig. 1: The authors show a clearly increased Matrigel plug vascularization, which is interesting, but also a bit puzzling - do Tregs contribute to Matrigel vascularization? Are there Tregs in the plug? Do they have more AREG?

Fig. 5: Does ischemia alone suffice to upregulate AREG and induce proangiogenic and anti-inflammatory effects in Tregs? Or does it have to be ischemic muscle lysate as shown in Fig. 6D?

Fig. 6: Please provide quantification of nuclear localization (and/or western blotting data after cell fractionation) - the evidence for increased nuclear localization in KO cells is currently rather anecdotal (one cell?!).

Fig. 7: If GPR174 inhibits EGR1-mediated AREG expression and if LysoPS is here the relevant GPR174 ligand - does LysoPS inhibit AREG expression (or EGR1-ChIP of AREG promoter sequences)?

Minor points:

- I am confused by the many P-values " $P > 0.9999$ " (e.g., Fig. 3C, Suppl Fig 5) - is this a copy/paste mistake? The values do not look like $p=1$.
- Abstract/Discussion: If the use of H89 in Fig. 7G is indeed the only evidence for a role of Gs/cAMP/PKA, I find the statement "demonstrate that GPR174 increases the activation of Gas/cAMP/PKA signaling pathway to inhibit the binding of Egr1 to the Arg1 promoter" too far fetched - the data only show that inhibition of PKA phenocopies knockdown of GPR174. Please tone down (for example, line 52, 336, line 339, lines 363-365 & 397-98). If the authors want to make this point, they should show effects of Gnas kd and/or altered cAMP levels after LysoPS stimulation in the relevant cell population.
- Fig. 1C: the arrangement within the bars ("2" sits between "0" and "1"?) and the colour coding (use red for the worst state?) could be more intuitive.
- Line 135: "muscle tissues of Foxp3+ mice" - I guess that should be FoxP3(GFP) mice?
- Lines 158 and 397: "Treg-expressed" seems a better term than "Treg-derived" (derived implies for me that the molecule is secreted/released from the respective cell).
- Fig. 3: Figure legend for I-K could be clearer (only reference to K, but not I,J)?
- Fig. 5F: Something is wrong with the labelling (+/- gefitinib). Also "porous membrane", not "poprous" (same in Supplemental Figure 5C).
- Please provide data availability statement for mRNAseq data.
- Some sentences require minor revision, e.g., L247. "Gating strategies was used...", L255: "two mice strains", L285 "regulates growth factors expression"
- Fig. S1: The term "Neovascularization" in the title of the figure legend is misleading, since we are here looking at basal vascularization.
- Fig. S4: Please indicate AREG in the volcano plot
- Line 171: "LysoPS levels" or "production" seems better than "LysoPS expression"

Response Letter

REVIEWER COMMENTS

Reviewer #1 (Remarks to the Author):

The manuscript by Liu et al. seeks to demonstrate the the G-coupled protein receptor GPR174 deletion increases blood flow recovery in mice following hindlimb ischemia. They go on to describe a, probably not the only, mechanism. I offer the following suggestions and comments to the authors.

1) The problem in PAD is impaired blood flow and the findings from the hind-limb ischemia model are that genetic deletion accelerates the rate of perfusion recovery over normal. The study does not report on impaired perfusion recovery or impaired angiogenesis. This occurs because the mice are quite young at the time of the HLI induction. Also, the authors used only male mice which by itself is not a limitation but added to the prior is more of an issue. Perhaps the authors could study the mice at more advanced age; add high fat diet or streptozocin. The other reason to make this point is the same mouse strain has been shown to have improved survival with a lesser cytokine response to LPS. This is clearly linked to an M1-like macrophage response and a greater M1-like response is linked to poorer perfusion recovery; making the major finding if not mechanism predictable.

2) GPR174 belongs to a G protein-coupled receptor superfamily which raises the question as to whether the genetic deletion results in changes in expression of the related GPR (GCPR, being the better abbreviation). An examination of a few related ones would be ideal.

3) Differences in the "early" response to HLI is often linked to macrophages. Are the macrophages from genetic deletion mice similar to the wild type, in the absence of HLI?

4) A number of the studies are done on mice Day post-HLI but at that time point the perfusion recovery in knock out vs. controls is already different. This raises the possibility that the differences in perfusion recovery are not the cause of the outcome difference but a result of the outcome difference. Redoing all the studies is not necessary but selecting a few outcomes at Day 3 (like comment 1) would strengthen the overall findings.

5) The wording on the transition from the HLI studies to the Matrigel needs to be modified. Most would consider HLI the in-vivo model and the matrigel an ex-vivo model. Perhaps what they mean is to begin to elucidate the mechanism of

the effects.

Reviewer #2 (Remarks to the Author):

In the current manuscript Liu et al. demonstrate that GPR174 negatively regulates neovascularization by inhibiting Treg function. They demonstrate that GPR174 deficiency significantly improves blood flow recovery and mitigates the inflammatory response in ischemic tissues. Furthermore, they show that GPR174 limits angiogenesis and vascular remodeling and promotes inflammation by inhibiting AREG expression. Mechanistic investigations demonstrate that GPR174 increases the activation of the Gas/cAMP/PKA signaling pathway to inhibit the binding of EGR1 to the Areg promoter.

As such this manuscript provides strong evidence that GPR174 inhibits neovascularization by downregulating AREG expression via the Gas/cAMP/PKA/EGR1 signaling pathway.

The manuscript is very well written, the experimental set-up and build-up of the arguments is excellent and provide a very complete set of data that support their conclusions.

I am impressed the thorough approach and completeness of the data set presented and have no further comments, except one minor point:

The authors report in the introduction : “ Tregs have also been reported to enhance the regeneration of peripheral vasculature in PAD12,13”, it would correct to refer also to the first two papers addressing a role for Tregs in the regulation of neovascularization :

1. Zougari Y, Ait-Oufella H, Waeckel L, et al. Regulatory T cells modulate postischemic neovascularization. *Circulation*. 2009;120:1415–25.
2. Hellingman AA, van der Vlugt LE, Lijkwan MA, Bastiaansen AJ, Sparwasser T, Smits HH, et al. . A limited role for regulatory T cells in post-ischemic neovascularization. *J Cell Mol Med*. (2012) 16:328–36. 10.1111/j.1582-4934.2011.01300.x

Reviewer #3 (Remarks to the Author):

This is a strong report by Lui and colleagues, and I congratulate the authors for this well-performed study. The authors claim that activation of regulatory T cells via a deletion of GPR174, an inhibitory receptor, leads to Treg activation and

production of Areg. This, in turn, promotes the neovascularization in mice after HLI. They showed that Areg indeed promotes endothelial cell functions, and the authors could block this convincingly with an sh-AAV vector system. In addition, they investigated the molecular footprint at the Areg gene locus and identified EGR1 as an initiator of Areg gene expression. This work is of high significance to the field of regenerative medicine and coagulation medicine, and it is overall well supported by statistical tests and advanced imaging and laboratory techniques. Although the importance of Treg in the field of ischemia and reperfusion injury is known, this study extends on the finding that Treg cells are also important after HLI. Again, the authors identify GPR174 as an inhibitory receptor, another fact already known, and the effects on myeloid populations (e.g. Qiu et al., „Gpr174-deficient regulatory T cells decrease cytokine storm in septic mice, “Nature 2019). This being said, the study is a well-performed investigation of the GPR174-coupled Treg function in HLI and is thus relevant for the field.

My remarks in details:

- In Figure 2, the authors convincingly show that Gpr174 mRNA is expressed in Treg cells upon Isch, but not Ctrl. Why do they not include other regeneration-related cell types such as ILC2s and myeloid cell populations? It would be great to add these to the figure and this would be a requirement for publication. Also, this is important to address Gpr174 in the future (as a therapeutic option). Is there a good flow cytometry antibody? That would help. Otherwise, sorting and mRNA are also good.
- All figures: please show standard deviation instead of SEM.
- All figures: please add % into the sorting gates, where shown (missing in Figure S2, for example)
- Line 127: please relabel „Figure 2L-N“ into „Figure 1L-N“
- Line 135, 145: what are Foxp3+ mice? Do you mean Foxp3(GFP) reporter animals? Please verify this, add the citation to identify these mice, and be cautious throughout the ms when you refer to sorted Foxp3+ Treg cells: if you sorted Foxp3(GFP)+ Treg cells, this makes sense; if you sorted Foxp3+ Treg cells, this would require fixation and permeabilization and sorting for RNA would be useless
- I do not understand Figure 5E: what is the difference between the left and right groups in each plot? Did you forget to add another discriminator (e.g. the GPR174 KO vs WT?)
- Figure 3A: I, personally, would also show the full RNA-seq results from the supplement and then highlight Areg and Egr1. Just a recommendation.

Reviewer #4 (Remarks to the Author):

Liu et al investigate in this study whether GPR174 affects ischemic disease by regulating Treg function. The authors provide evidence that GPR174 deficiency in Tregs results in reduced inflammation and improved endothelial function and suggest that this is due to a signalling cascade involving GPR174/PKA-dependent EGR1 translocation with consecutive upregulation of AREG expression.

This is a very thorough and well-written study, the topic is interesting and relevant, the methods are state-of-the art. I have only a few major points and some minor issues:

Major points:

Fig. 1: The authors show a clearly increased Matrigel plug vascularization, which is interesting, but also a bit puzzling - do Tregs contribute to Matrigel vascularization? Are there Tregs in the plug? Do they have more AREG?

Fig. 5: Does ischemia alone suffice to upregulate AREG and induce proangiogenic and anti-inflammatory effects in Tregs? Or does it have to be ischemic muscle lysate as shown in Fig. 6D?

Fig. 6: Please provide quantification of nuclear localization (and/or western blotting data after cell fractionation) - the evidence for increased nuclear localization in KO cells is currently rather anecdotal (one cell?!).

Fig. 7: If GPR174 inhibits EGR1-mediated AREG expression and if LysoPS is here the relevant GPR174 ligand – does LysoPS inhibit AREG expression (or EGR1-ChIP of AREG promoter sequences)?

Minor points:

- I am confused by the many P-values “P>0.9999” (e.g., Fig. 3C, Suppl Fig 5) – is this a copy/paste mistake? The values do not look like p=1.

- Abstract/Discussion: If the use of H89 in Fig. 7G is indeed the only evidence for a role of Gs/cAMP/PKA, I find the statement “demonstrate that GPR174 increases the activation of Gas/cAMP/PKA signaling pathway to inhibit the binding of Egr1 to the Arg1 promoter” too far fetched – the data only show that inhibition of PKA phenocopies knockdown of GPR174. Please tone down (for example, line 52, 336, line 339, lines 363-365 & 397-98). If the authors want to make this point, they should show effects of Gnas kd and/or altered cAMP levels after LysoPS stimulation in the relevant cell population.

- Fig. 1C: the arrangement within the bars (“2” sits between “0” and “1”?) and the colour coding (use red for the worst state?) could be more intuitive.

- Line 135: “muscle tissues of Foxp3+ mice” – I guess that should be FoxP3(GFP) mice?

- Lines 158 and 397: “Treg-expressed” seems a better term than “Treg-derived” (derived implies for me that the molecule is secreted/released from the respective cell).

- Fig. 3: Figure legend for I-K could be clearer (only reference to K, but not I,J)?
- Fig. 5F: Something is wrong with the labelling (+/- gefitinib). Also “porous membrane”, not “poprous” (same in Supplemental Figure 5C).
- Please provide data availability statement for mRNAseq data.
- Some sentences require minor revision, e.g., L247. “Gating strategies was used...”, L255: “two mice strains”, L285 “regulates growth factors expression”
- Fig. S1: The term “Neovascularization” in the title of the figure legend is misleading, since we are here looking at basal vascularization.
- Fig. S4: Please indicate AREG in the volcano plot
- Line 171: “LysoPS levels” or “production” seems better than “LysoPS expression”

Response to Reviewers

We are grateful to the Reviewers for their time and thoughtful suggestions. We have appropriately revised our manuscript by including data from new experiments while also addressing all the comments and suggestions. Reviewer's comments are in **black/bold**, our response in blue.

Reviewer #1 (Remarks to the Author):

The manuscript by Liu et al. seeks to demonstrate the the G-coupled protein receptor GPR174 deletion increases blood flow recovery in mice following hindlimb ischemia. They go on to describe a, probably not the only, mechanism. I offer the following suggestions and comments to the authors.

1) The problem in PAD is impaired blood flow and the findings from the hind-limb ischemia model are that genetic deletion accelerates the rate of perfusion recovery over normal. The study does not report on impaired perfusion recovery or impaired angiogenesis. This occurs because the mice are quite young at the time of the HLI induction. Also, the authors used only male mice which by itself is not a limitation but added to the prior is more of an issue. Perhaps the authors could study the mice at more advanced age; add high fat diet or streptozocin. The other reason to make this point is the same mouse strain has been shown to have improved survival with a lesser cytokine response to LPS. This is clearly linked to an M1-like macrophage response and a greater M1-like response is linked to poorer perfusion recovery; making the major finding if not mechanism predictable.

Response: Thank you for these insightful comments. Following efforts are made to address mentioned issues now.

(1) According to comments from the reviewer and the editor, hindlimb ischemic model was now established in streptozotocin (STZ)-induced diabetic mice to

study the role of GPR174 in pathological condition. Briefly, six-week-old-male GPR174^{-Y} mice and WT littermates were fasted for 6h and then injected intraperitoneally (i.p.) with citrate buffer or STZ (55 mg/kg in sodium citrate buffer, pH 4.5) for 5 consecutive days. One week later, blood glucose concentrations were measured, and only mice with non-fasting blood glucose levels > 280 mg/dL were considered diabetic and used for experiments. Surgical intervention to induce unilateral hindlimb ischemia was performed 12 weeks after constant hyperglycemia (**Supplementary Fig. 13a, b**). The results showed that GPR174 knockout enhances blood flow recovery after HLI in diabetic mice. The results are shown in **Supplementary Fig. 13c, d, e, f**, and discussed in the revised manuscript (**page 16-17, lines 329-342**).

Supplementary Fig. 13. GPR174 knockout enhances blood flow recovery after HLI in diabetic mice. (a) A schematic model of streptozotocin (STZ)-induced diabetic mouse ischemic hindlimb model. (b) Blood glucose levels in STZ-induced diabetic mice (n=10-13). (c, d) Representative images and quantification of hindlimb blood perfusion in WT and GPR174^{-/-} diabetic mice (n=5-8). (e, f) Representative immunofluorescent images of CD31 staining and

quantification of CD31⁺ area in diabetic mice gastrocnemius cross sections at the indicated times after HLI (n=4). (g, h) Representative immunofluorescent images of the proliferation and apoptosis of ECs 3 days after HLI (n=5). Scale bars, 50 μ m.

The study does not report on impaired perfusion recovery or impaired angiogenesis.

Thanks for the valuable comments. We performed additional hindlimb ischemia experiments on diabetic mice and evaluated blood flow perfusion in this model. The results showed that perfusion recovery impairment post hindlimb ischemia was more significant in diabetic mice as compared with non-diabetic controls. GPR174 knockout also improved perfusion recovery under diabetic condition. The results are shown in **Supplementary Fig. 13c, d**.

Also, the authors used only male mice which by itself is not a limitation but added to the prior is more of an issue.

Thanks for this comment. Yes. We agree. In our study design, we only considered to exclude the impact of sex hormone on blood flow recovery in our model. We added this limitation in the revised manuscript and planned to perform similar studies in female mice to address this issue in the near future. It reads (**page 20, lines 415-419**): "It is to note that in our study, we only studied the impact of GPR174 knockout in male mice undergoing HLI, to minimize the

influence from sex hormone variations from female mice, caution is thus needed in the interpretation of our results, and future studies are needed to explore the impact of GPR174 knockout in female mice in identical experimental conditions.”

The other reason to make this point is the same mouse strain has been shown to have improved survival with a lesser cytokine response to LPS. This is clearly linked to an M1-like macrophage response and a greater M1-like response is linked to poorer perfusion recovery; making the major finding if not mechanism predictable.

Thanks a lot for pointing out this important issue. We agree and added following description in the revised manuscript now (**page 19-20, lines 395-404**): “Our results showed that compared with wild-type Tregs, GPR174-deficient Tregs promoted anti-inflammatory polarization of macrophages after HLI, which was in line with previous finding by Dongze Qiu and colleagues in that GPR174 KO mice from the same mouse strain attenuated LPS induced inflammation response in macrophage (PMID: 30850582). In addition, they also showed that GPR174-deficient Tregs secreted IL-10, leading to increased M2 polarization of macrophages, thereby protecting the mice against sepsis induced lung injury (PMID: 30850582). In our study, IL-10 was similar between wild-type and GPR174-deficient Tregs after hindlimb ischemia, the difference might possibly be related to the degree of inflammation response, in that inflammation response is overwhelming in sepsis model, while ischemia is dominant in our HLI model.”

2) GPR174 belongs to a G protein-coupled receptor superfamily which raises the question as to whether the genetic deletion results in changes in expression of the related GPR (GPCR, being the better abbreviation). An examination of a few related ones would be ideal.

Response: Thanks and we followed the helpful suggestion. GPCR was used in the revised manuscript now. To identify the potential influence of genetic

GPR174 deletion on other related G protein-coupled receptors, we referred STRING (<https://string-db.org/>) to search the potential related GPCR that might interact with GPR174. The results showed that the related G protein-coupled receptors included GPR61, GPR114 and GPR139. Meanwhile, previous study also reported that GPR34, P2RY10 and GPR174 comprise a LysoPS receptor family (PMID: 22983457). Thus, we detected the expression of the above GPCRs in Tregs at the mRNA level post HLI, and the results indicated that GPR174 deletion did not affect their expression. Please see the **Review Figure 1**.

Review Figure 1. (a) STRING analysis of GPR174. (b) The related GPCRs mRNA expression in Tregs post HLI (n=4).

3) Differences in the "early" response to HLI is often linked to macrophages. Are the macrophages from genetic deletion mice similar to the wild type, in the absence of HLI?

Response: Thanks for the valuable comments. To address the issue, macrophages were detected by flow cytometry. Results showed that macrophages in the non-ligated tissue were similar between wild type and GPR174-deficient mice (**Supplementary Fig. 9b, c**)

Supplementary Fig. 9b, c. Representative flow cytometric dot plots to determine neutrophils, macrophages in the non-ligated gastrocnemius of WT and GPR174^{-/-} mice (n=6).

4) A number of the studies are done on mice Day post-HLI but at that time point the perfusion recovery in knock out vs. controls is already different. This raises the possibility that the differences in perfusion recovery are not the cause of the outcome difference but a result of the outcome difference. Redoing all the studies is not necessary but selecting a few outcomes at Day 3 (like comment 1) would strengthen the overall findings.

Response: Thank you for raising this thoughtful comment. We performed additional experiments to address these issues.

As suggested, we detected reperfusion and angiogenesis in non-diabetic and diabetic mice.

The new results are updated in the revised manuscript. **Page 11, lines 204-211:** “Perfusion recovery in GPR174^{-/-} vs. controls might already be different early after HLI. To address this issue, we assessed reperfusion and angiogenesis at 3 days post HLI. Results showed reperfusion and angiogenesis were similar between WT and GPR174^{-/-} mice at 3 days after HLI (**Fig. 1a, b, d, e**). Further, immunofluorescence was performed to detect the proliferation and apoptosis of ECs after HLI. Elevated proliferation and reduced apoptosis

of ECs were evidenced in GPR174 KO mice at 3 days after HLI (Supplementary Fig. 7a, b), suggesting that GPR174 knockout promotes blood flow recovery by affecting endothelial cell proliferation and apoptosis.”

Fig. 1a, b, d, e. (a, b) Representative images and quantification of hindlimb blood perfusion in WT and GPR174^{-/-} mice using laser Doppler imaging at the indicated times after HLI (n=15-16). (d, e) Representative immunofluorescent images of CD31 staining and quantification of CD31⁺ area in WT and GPR174^{-/-} mice gastrocnemius cross sections at the indicated times after HLI (n=5-6). Scale bar, 50 μm.

Supplementary Fig. 7a, b. (a, b) apoptosis and proliferation of ECs on day 3 after HLI. scale bars, 50 μm.

In addition, we detected reperfusion and angiogenesis in STZ-induced diabetic mice. Reperfusion and angiogenesis in WT and GPR174 KO mice showed no difference 3 days after HLI (**Supplementary Fig. 13c-f**). Further, immunofluorescence was performed to detect the proliferation and apoptosis of ECs after HLI. ECs in GPR174 KO group showed elevated proliferation and reduced apoptosis (**Supplementary Fig. 13g, h**). Seven days after HLI, blood flow recovery was more pronounced in GPR174^{-/-} diabetic mice (**Supplementary Fig. 13c, d**). These results indicated that GPR174 does play a role in endothelial cell proliferation and apoptosis as early as 3 days post HLI. The increased proliferation and decreased apoptosis of ECs by GPR174 KO might partially be associated to the observed promoted perfusion recovery in later days post HLI in our study.

Supplementary Fig. 13c-h. (c, d) Representative images and quantification of hindlimb blood perfusion in WT and GPR174^{-/-} diabetic mice (n=5-8). (e, f) Representative immunofluorescent images of CD31 staining and quantification of CD31⁺ area in diabetic mice gastrocnemius cross sections at the indicated times after HLI (n=4). (g, h) Representative immunofluorescent images of the proliferation and apoptosis of ECs 3 days after HLI (n=5). Scale bars, 50 μ m.

5) The wording on the transition from the HLI studies to the Matrigel needs to be modified. Most would consider HLI the in-vivo model and the matrigel an ex-vivo model. Perhaps what they mean is to begin to

elucidate the mechanism of the effects.

Response: Thanks so much for your help. We fully agree. The manuscript is modified now (**page 6, lines 107-109**). “In addition, the Matrigel plug assay, an ex vivo model, was used to further explore the potential impact of GPR174 deletion in angiogenesis.”

Reviewer #2 (Remarks to the Author):

In the current manuscript Liu et al. demonstrate that GPR174 negatively regulates neovascularization by inhibiting Treg function. They demonstrate that GPR174 deficiency significantly improves blood flow recovery and mitigates the inflammatory response in ischemic tissues. Furthermore, they show that GPR174 limits angiogenesis and vascular remodeling and promotes inflammation by inhibiting AREG expression. Mechanistic investigations demonstrate that GPR174 increases the activation of the G α s/cAMP/PKA signaling pathway to inhibit the binding of EGR1 to the Areg promoter.

As such this manuscript provides strong evidence that GPR174 inhibits neovascularization by downregulating AREG expression via the G α s/cAMP/PKA/EGR1 signaling pathway.

The manuscript is very well written, the experimental set-up and build-up of the arguments is excellent and provide a very complete set of data that support their conclusions.

I am impressed the thorough approach and completeness of the data set presented and have no further comments, except one minor point:

The authors report in the introduction : “ Tregs have also been reported to enhance the regeneration of peripheral vasculature in PAD12,13”, it would correct to refer also to the first two papers addressing a role for Tregs in the regulation of neovascularization :

- 1. Zougari Y, Ait-Oufella H, Waeckel L, et al. Regulatory T cells modulate postischemic neovascularization. *Circulation*. 2009;120:1415–25.**
- 2. Hellingman AA, van der Vlugt LE, Lijkwan MA, Bastiaansen AJ, Sparwasser T, Smits HH, et al. . A limited role for regulatory T cells in post-ischemic neovascularization. *J Cell Mol Med*. (2012) 16:328–36.**

10.1111/j.1582-4934.2011.01300.x

Response: Thanks so much for the positive comments and suggestion on our manuscript. We have added the two landmark studies (PMID: 19770391, 21426486) in the introduction section of the revised manuscript (**page 4, lines 53-54**).

Reviewer #3 (Remarks to the Author):

This is a strong report by Lui and colleagues, and I congratulate the authors for this well-performed study. The authors claim that activation of regulatory T cells via a deletion of GPR174, an inhibitory receptor, leads to Treg activation and production of Areg. This, in turn, promotes the neovascularization in mice after HLI. They showed that Areg indeed promotes endothelial cell functions, and the authors could block this convincingly with an sh-AAV vector system. In addition, they investigated the molecular footprint at the Areg gene locus and identified EGR1 as an initiator of Areg gene expression. This work is of high significance to the field of regenerative medicine and coagulation medicine, and it is overall well supported by statistical tests and advanced imaging and laboratory techniques. Although the importance of Treg in the field of ischemia and reperfusion injury is known, this study extends on the finding that Treg cells are also important after HLI. Again, the authors identify GPR174 as an inhibitory receptor, another fact already known, and the effects on myeloid populations (e.g. Qiu et al., „Gpr174-deficient regulatory T cells decrease cytokine storm in septic mice, “Nature 2019). This being said, the study is a well-performed investigation of the GPR174-coupled Treg function in HLI and is thus relevant for the field.

Major

- In Figure 2, the authors convincingly show that Gpr174 mRNA is expressed in Treg cells upon Isch, but not Ctrl. Why do they not include other regeneration-related cell types such as ILC2s and myeloid cell populations? It would be great to add these to the figure and this would be a requirement for publication. Also, this is important to address Gpr174 in the future (as a therapeutic option). Is there a good flow cytometry antibody? That would help. Otherwise, sorting and mRNA are also good.

Response: Thank you so much for raising these important points.

(1) According to the valuable suggestions, we detected GPR174 expression in CD45⁺CD3⁻B220⁻CD90.2⁺KLRG1⁺ILC2s, CD45⁺CD11b⁺Ly6G⁺neutrophils, and CD45⁺CD11b⁺Ly6G⁻F4/80⁺macrophages. The results showed that the expression of GPR174 remained unchanged in macrophages, neutrophils, ILC2s, B cells, CD8⁺ T cells, and CD4⁺Foxp3⁻ T cells, but upregulated in Tregs after HLI (**Fig. 2a**).

Fig. 2. GPR174-deficient Tregs improve blood flow recovery after HLI. (a) GPR174 mRNA expression in B cells, CD8⁺ T cells, CD4⁺Foxp3⁻ T cells, CD4⁺Foxp3^{GFP} Tregs, ILC2s, Neutrophils, and macrophages isolated from non-ischemic and ischemic muscle in WT mice (n=4).

(2) We tried but could not find suitable flow cytometry antibody for GPR174. Therefore, we used cell sorting and detected mRNA expression in our study as suggested.

Minor

- All figures: please show standard deviation instead of SEM.

Response: Thank you for the suggestion. As suggested, in the revised manuscript, we presented all statistical results with standard deviation (SD) now.

- All figures: please add % into the sorting gates, where shown (missing

in Figure S2, for example)

Response: Thanks for the helpful suggestion. We added % into all sorting gates now, as suggested (**Supplementary Fig. 2a and 7a**).

- **Line 127: please relabel „Figure 2L-N“ into „Figure 1L-N“**

Response: We apologize for the typo. We relabeled “Figure 2L-N” into “Figure 1L-N” in the revised manuscript now (**page 6, lines 110**).

- **Line 135, 145: what are Foxp3⁺ mice? Do you mean Foxp3(GFP) reporter animals? Please verify this, add the citation to identify these mice, and be cautious throughout the ms when you refer to sorted Foxp3⁺ Treg cells: if you sorted Foxp3(GFP)⁺ Treg cells, this makes sense; if you sorted Foxp3⁺ Treg cells, this would require fixation and permeabilization and sorting for RNA would be useless**

Response: Thank you for these insightful comments. In our manuscript, Foxp3^{GFP} mice were inadvertently written as Foxp3⁺ mice. And we have added related references (PMID: 15780990) to identify Foxp3^{GFP} mice (**page 7, lines 119**). In our study, CD4⁺Foxp3^(GFP)⁺Tregs were sorted from Foxp3^{GFP} mice or GPR174^{-/-};Foxp3^{GFP} mice and these points are clarified in the revised manuscript. Thanks for help.

- **I do not understand Figure 5E: what is the difference between the left and right groups in each plot? Did you forget to add another discriminator (e.g. the GPR174 KO vs WT?)**

Response: Sorry for the confusion. We have corrected the legend now.

- Figure 3A: I, personally, would also show the full RNA-seq results from the supplement and then highlight Areg and Egr1. Just a recommendation.

Response: Thank you for addressing this important point. As suggested, we have added annotations to highlight Areg and Egr1 in the volcano plot (Supplementary Fig. 4a).

Supplementary Figure 4. RNA-seq analysis of gene expression profile in muscles from WT controls and GPR174^{-/-} mice post HLI. (a) Volcano plot of differentially expressed genes in muscles of GPR174KO compared with wild-type 7 days post HLI (n=3).

Reviewer #4 (Remarks to the Author):

Liu et al investigate in this study whether GPR174 affects ischemic disease by regulating Treg function. The authors provide evidence that GPR174 deficiency in Tregs results in reduced inflammation and improved endothelial function and suggest that this is due to a signalling cascade involving GPR174/PKA-dependent EGR1 translocation with consecutive upregulation of AREG expression.

This is a very thorough and well-written study, the topic is interesting and relevant, the methods are state-of-the art. I have only a few major points and some minor issues:

Major points:

Fig. 1: The authors show a clearly increased Matrigel plug vascularization, which is interesting, but also a bit puzzling - do Tregs contribute to Matrigel vascularization? Are there Tregs in the plug? Do they have more AREG?

Response: Thanks very much for point out this important issue.

(1) A previous study (PMID: 21753853) has shown that supernatants from hypoxic mouse CD4⁺CD25⁺ splenocytes promote angiogenesis in Matrigel plugs. We thus investigated the effect of Tregs on angiogenesis in Matrigel plugs after in vivo depletion with an anti-CD25-specific antibody, the PC61 mAb (anti-murine CD25 rat IgG1), which is widely used for depletion of Tregs (PMID: 20039297, 24966183), and rat IgG isotype was used as the control treatment. Results showed that hemoglobin levels were decreased in Matrigel plugs in the PC61 antibody injection mice compared with IgG isotype injection mice, indicating Tregs contribute to Matrigel vascularization. Please see the **Review Figure 2**.

Review Figure 2. (a) Hemoglobin levels in the Matrigel plugs. (b) immunofluorescence staining of transverse sections of the Matrigel plugs (DAPI [blue]; endothelium marker isolectin B4 [green]); scale bars, 50 μ m.

(2) We detected Tregs infiltration in Matrigel plugs by flow cytometry. The results showed that Tregs were infiltrated in both GPR174-deficient and control groups. But no significant difference was detected between the two groups. This result is shown in **Supplementary Fig. 6a** now.

(3) We followed the helpful suggestion, and further evaluated AREG expression. Since direct-labeled flow cytometric antibody, ELISA was used to detect the content of AREG in Matrigel plugs. The results indicated that AREG levels in GPR174-deficient Tregs were higher than that in control group in Matrigel plugs. (**Supplementary Fig. 6b**)

Supplementary Fig. 6. GPR174-deficient Tregs promote angiogenesis via AREG in Matrigel plugs.

Fig. 5: Does ischemia alone suffice to upregulate AREG and induce proangiogenic and anti-inflammatory effects in Tregs? Or does it have to

be ischemic muscle lysate as shown in Fig. 6D?

Response: Thanks for the meaningful comments. Our study demonstrated that ischemia is sufficient to elevate AREG secretion, induce enhanced proangiogenic and anti-inflammatory effects in GPR174-deficient Tregs compared with wild-type groups, as shown in the Fig. 3c, and 5d.

Fig. 3c. Relative mRNA levels of Areg, Il-10, and Vegf in Tregs sorted from the gastrocnemius tissues of WT and GPR174^{-/-} mice (n=4).

Fig. 5d. Relative mRNA levels of proinflammatory (upper panel) and anti-inflammatory genes in macrophages co-cultured with Treg cells and anti-AREG antibody for 24 hours (n=4).

The manuscript is updated now accordingly (Page 15, lines 293-295). “To simulate the ischemic environment in vitro, Tregs isolated from spleen of

GPR174^{-Y} and wild-type mice were treated with shEgr1 (Supplementary Fig. 10) and cultured with ischemic muscle lysates.”

Fig. 6: Please provide quantification of nuclear localization (and/or western blotting data after cell fractionation) - the evidence for increased nuclear localization in KO cells is currently rather anecdotal (one cell?!).

Response: Thanks so much for the helpful comments. We have increased the number of cells per field and have quantified the nuclear localization now and results are presented in **Fig 6c, d**.

Fig. 6c, d. (c) Representative immunofluorescent images of EGR1 (red), and DAPI (blue) staining in Tregs isolated from the ischemic muscles of WT and GPR174^{-Y} mice 7 days after HLI. Scale bar, 10 μ m. (d) the quantification of nuclear localization of Egr1 (n=5).

Fig. 7: If GPR174 inhibits EGR1-mediated AREG expression and if LysoPS is here the relevant GPR174 ligand – does LysoPS inhibit AREG expression (or EGR1-ChIP of AREG promoter sequences)?

Response: Thanks so much for this comment. To address the concern, Tregs were isolated from the spleen of wild-type mice and stimulated with LysoPS. AREG expression was evaluated by qPCR and ELISA. Results showed that LysoPS inhibited AREG expression in Tregs, this new result is shown in **Supplementary Fig. 11a, b** now.

Supplementary Fig. 11a, b. AREG expression in Tregs treated with LysoPS, db-cAMP, Rp-cAMPS, and H89 (n=4).

Minor points:

- I am confused by the many P-values “P>0.9999” (e.g., Fig. 3C, Suppl Fig 5) – is this a copy/paste mistake? The values do not look like p=1.

Response: Thank you for the comment. We reanalyzed the data and corrected the P-values of Fig. 3c and Supplementary Fig. 5a, b now.

Fig. 3c. Relative mRNA levels of Areg, Il-10, and Vegf in Tregs sorted from the gastrocnemius tissues of WT and GPR174^{-/-} mice (n=4).

Supplementary Fig. 5. (a) Relative mRNA levels of AREG in gastrocnemius from wild-type and GPR174^{-/-} mice 7 days post HLI (n=8). (b) Serum AREG, IL-10, and VEGF protein content in wild-type and GPR174^{-/-} mice 7 days post HLI (n=9-16).

- **Abstract/Discussion:** If the use of H89 in Fig. 7G is indeed the only evidence for a role of Gs/cAMP/PKA, I find the statement “demonstrate that GPR174 increases the activation of Gas/cAMP/PKA signaling pathway to inhibit the binding of Egr1 to the Arg1 promoter” too far fetched – the data only show that inhibition of PKA phenocopies knockdown of GPR174. Please tone down (for example, line 52, 336, line 339, lines 363-365 & 397-98). If the authors want to make this point, they should show effects of Gnas kd and/or altered cAMP levels after LysoPS stimulation in the relevant cell population.

Response: Thank you for this fair comment. We performed additional experiments to clarify this issue.

According to the suggestion, Tregs were isolated from murine spleen and stimulated with LysoPS, LysoPS+db-cAMP, LysoPS+Rp-cAMPS, and

LysoPS+H 89, and then Areg levels were assessed by RT-PCR and ELISA. We found that Areg expression was affected by altered cAMP or PKA levels in Tregs after LysoPS treatment, the results were summarized in **Supplementary Fig. 11a, b** and contents were updated in the revised manuscript accordingly (**page 15, lines 296-300**). “We then investigated how LysoPS regulates AREG expression. AREG production was suppressed in Tregs treated with LysoPS and the cell-permeable cAMP analog db-cAMP (Supplementary Fig. 11a, b). In contrast, blocking cAMP-dependent PKA with Rp-cAMPS or H-89 reversed the effects of LysoPS (Supplementary Fig. 11a, b).”

Supplementary Fig. 11. AREG expression in Tregs treated with LysoPS, db-cAMP, Rp-cAMPS, and H89. (n=4).

Fig. 1C: the arrangement within the bars (“2” sits between “0” and “1”?) and the colour coding (use red for the worst state?) could be more intuitive.

Response: Thanks so much. As suggested, we have re-organized the **Fig. 1c** with color coding to enhance the readability.

- Line 135: “muscle tissues of Foxp3+ mice” – I guess that should be FoxP3(GFP) mice?

Response: We apologize for the typo. In our study, Tregs were sorted from Foxp3^{GFP} mice. The term of Foxp3^{GFP} mice was used in the revised manuscript (page 7, lines 119).

- Lines 158 and 397: “Treg-expressed” seems a better term than “Treg-derived” (derived implies for me that the molecule is secreted/released from the respective cell).

Response: Thanks for the help. As suggested, in the revised manuscript, we have replaced the term “Treg-derived” with “Treg-expressed”. (Page 7, lines 127; page 8, lines 142,144; page 21, lines 423)

- Fig. 3: Figure legend for I-K could be clearer (only reference to K, but not I,J)?

Response: Thanks. We have rephrased the Figure legend as suggested (page 45, Lines 870-874). “i, j Representative immunofluorescent images of CD31 (i) and α SMA (k) staining and quantification of CD31 (j) and lumen perimeter (l) in muscle cross sections (n=6). Scale bar, 50 μ m. k, l Representative immunofluorescent images α SMA (k) staining and quantification of lumen perimeter (l) in muscle cross sections (n=6). Scale bar, 50 μ m.”

- Fig. 5F: Something is wrong with the labelling (+/- gefitinib). Also “porous membrane”, not “poprous” (same in Supplemental Figure 5C).

Response: Thanks. As suggested, we have added “Vehicle (red box) / Gefitinib (blue box)” to the legend on the old Fig. 5e and corrected the misspelled word “poprous” to “porous” on Fig. 5c and Supplementary Fig. 5c; please see Fig. 5c, e and Supplementary Fig. 5c.

Fig. 5c, e. (c) Scheme of Tregs-macrophages co-culture. (e) Quantification of bioactive TGF-β and VEGF in macrophages stimulated with recombinant AREG in the presence or absence of inhibitors for the EGFR (Gefitinib) for 24 hours (n=8).

- Please provide data availability statement for mRNAseq data.

Response: Thanks. As suggested, in the revised manuscript, we have added data availability statement now (page 32, lines 677-680).

- Some sentences require minor revision, e.g., L247. “Gating strategies was used...”, L255: “two mice strains”, L285 “regulates growth factors expression”

Response: Thank you for your helpful suggestion. “Gating strategies was used” is replaced with “Gating strategies were used”; “two mice strains” is replaced with “two groups”; “regulates growth factors expression” is replaced with “a known transcription factor of growth factor” in the revised manuscript now. (Page 12, lines 243; page 13, lines 252; page 14, lines 280-281).

- Fig. S1: The term “Neovascularization” in the title of the figure legend is misleading, since we are here looking at basal vascularization.

Response: Thanks for the fair comment, we fully agree. As suggested, we have replaced “Global GPR174 deficiency has no effects on neovascularization” with “Global GPR174 deficiency has no effects on basal vascularization” in the legend title of **Supplementary Fig. 1**.

- Fig. S4: Please indicate AREG in the volcano plot

Response: Thanks for the helpful comment. As suggested, we have added a marker for Areg in the volcano plot now (**Supplementary Fig. 4a**).

- Line 171: “LysoPS levels” or “production” seems better than “LysoPS expression”

Response: Thanks and we followed the helpful suggestion. Now, we have replaced “LysoPS expression” with “LysoPS production” in the revised manuscript as suggested (**page 8, lines 155**).

Reviewer #1 (Remarks to the Author):

The manuscript has been extensively revised.

In the revised manuscript, Line 52 or 53 (new text) the sentence begins with "and" -- while there are situations for which that is appropriate, that is not the case here. That sentence should be edited.

The presentation of the p values should be altered. They should be rounded off and not be presented in to the ten-thousands (or greater). Where levels of significance were set at 0.05 anything greater should be p=NS.

Reviewer #3 (Remarks to the Author):

All my comments have been addressed.

Congratulations to this strong and relevant study!

Reviewer #4 (Remarks to the Author):

All my questions have been answered.

Response Letter

REVIEWER COMMENTS

Reviewer #1 (Remarks to the Author):

The manuscript has been extensively revised.

In the revised manuscript, Line 52 or 53 (new text) the sentence begins with "and" - while there are situations for which that is appropriate, that is not the case here. That sentence should be edited.

Response: Thank you for this comment. We have replaced "And other studies have revealed that Tregs play a role in angiogenesis after HLI." with "Conversely, other studies have revealed that depletion of Tregs exhibited increased restoration of blood flow after HLI." in the revised manuscript now. (Page 3, Lines 54-56)

The presentation of the p values should be altered. They should be rounded off and not be presented in to the ten-thousands (or greater). Where levels of significance were set at 0.05 anything greater should be p=NS.

Response: Thank you for your suggestion. As suggested, the presentation of the p values has been altered in the revised manuscript.

Reviewer #3 (Remarks to the Author):

All my comments have been addressed.

Congratulations to this strong and relevant study!

Response: We are grateful that the reviewer is fully satisfied of our study.

Reviewer #4 (Remarks to the Author):

All my questions have been answered.

Response: We appreciate the time and comments from the referee.